# Electrophysiological population dynamics reveal context dependencies during decision making in human frontal cortex

Wan-Yu Shih[1] ✉, Hsiang-Yu Yu[2,3,4], Cheng-Chia Lee[3,4,5], Chien-Chen Chou[2,3,4], Chien Chen[2,3,4], Paul W. Glimcher [6,7] ✉ & Shih-Wei Wu [1,4] ✉

Evidence from monkeys and humans suggests that the orbitofrontal cortex (OFC) encodes the subjective value of options under consideration during choice. Data from non-human primates suggests that these value signals are context-dependent, representing subjective value in a way influenced by the decision makers' recent experience. Using electrodes distributed throughout cortical and subcortical structures, human epilepsy patients performed an auction task where they repeatedly reported the subjective values they placed on snack food items. High-gamma activity in many cortical and subcortical sites including the OFC positively correlated with subjective value. Other OFC sites showed signals contextually modulated by the subjective value of previously offered goods—a context dependency predicted by theory but not previously observed in humans. These results suggest that value and value-context signals are simultaneously present but separately represented in human frontal cortical activity.

Over the course of the last several decades, studies in macaque monkeys have come to define the electrophysiological representation of rewards and reinforcers[1–11] (refs. 12–18 for reviews). This research has revealed that the firing rates of neurons in many brain areas encode a subjective estimate, the subjective value, of reward magnitude and type. Based on these extensive recordings, the broad topography of the network that represents reward-related value has now been well established in the macaque brain. Similar data are emerging for the rodent brain[19–25], further extending our understanding of these important electrophysiological signals.

One key feature of this work in animals is that it has revealed the importance of context in the subjectivization of these reward-related signals. Very broadly, context can be seen as a general category for describing the impact of the environment—most notably its spatial and temporal profiles—on brain activity and behavior. The process that leads to subsequent changes in behavior and brain activity is often referred to as adaptation and adaptive coding respectively[26–31]. Monkey parietal cortex, for example, has been shown to encode a kind of spatial context-dependency where the subjective value of an option presented in one spatial location impacts the neural value signals presented in another location[32–34]. Monkey orbitofrontal cortex (OFC) and dorsal anterior cingulate cortex (ACC), in contrast, appear to show a kind of temporal context-dependency, in which the recent history of rewards influences the electrophysiological representation of currently available rewards[15,17,28,35–37]. Closely related work has extended these findings to rodents[38]. The importance of these findings, however, extends beyond the study of non-human animals because growing evidence suggests that these subjectivized representations seem to account for important idiosyncrasies and irrationalities observed in human choice behavior[39–44].

At a neurobiological level, functional magnetic resonance imaging (fMRI) studies in humans have also provided links to animal-based

[1]Institute of Neuroscience, College of Life Sciences, National Yang Ming Chiao Tung University, Taipei, Taiwan, ROC. [2]College of Medicine, National Yang Ming Chiao Tung University, Taipei, Taiwan, ROC. [3]Department of Epilepsy, Neurological Institute, Taipei Veterans General Hospital, Taipei, Taiwan, ROC. [4]Brain Research Center, National Yang Ming Chiao Tung University, Taipei, Taiwan, ROC. [5]Department of Neurosurgery, Neurological Institute, Taipei Veterans General Hospital, Taipei, Taiwan, ROC. [6]Neuroscience Institute, NYU Grossman School of Medicine, New York, NY, USA. [7]Department of Neuroscience and Physiology, NYU Grossman School of Medicine, New York, NY, USA. ✉e-mail: wanyushih.y@nycu.edu.tw; paulg@nyu.edu; swwu@nycu.edu.tw

studies of the subjective value network to our understanding of the human brain. While the blood-oxygen-level-dependent (BOLD) signal measured by fMRI is quite distinct from electrophysiological measurements, many fMRI studies now show clear evidence of a subjective value network[45–47] similar to the one observed electrophysiologically in animals. Interestingly however, BOLD signal maps of subjective value in humans do not always agree with the electrophysiological maps developed in animals. In the parietal and orbitofrontal cortices, for example, very few studies using fMRI have identified the subjective value signals so often seen in non-human primate brains (though see ref. 48 for an important and perhaps explanatory exception).

When considering fMRI data, however, much less information is available about the role of context in the human neural representation of reward. Although a large amount of behavioral evidence identifies both spatial and temporal context-dependency as a critical factor that shapes both human and animal choice behavior[31,49–52], few studies exist that localize context dependency during decision-making in the human brain. Most studies to date showed context-dependent neural responses either when participants experienced an outcome (e.g., monetary gain or loss) or were presented with reward-predicting cues[30,53–56]. These context-dependent representations were observed in the standard subjective-value network including the ventromedial prefrontal cortex (vmPFC), striatum, and/or OFC. However, unlike in macaques, it is unclear in humans whether and how the OFC participates in context-dependent valuation during decision-making[29,57]. For example, at the time of choice, some brain regions in the subjective-value network assessed with fMRI would show context-dependent responses (ventral striatum) while others do not (vmPFC)[29]. This lack of clarity may reflect either a limitation of the technology or a species difference. While there are some examples of context-dependent responses in humans, it seems likely that the spatial and temporal scale at which fMRI operates and the nature of the BOLD signal itself, have made it extremely difficult to extract clear evidence of either spatial or temporal context dependency using that technology.

In this report, we sought to achieve three goals aimed at addressing these gaps between our understanding of human and animal representations of reward and reinforcement. First, and most importantly, we sought to determine whether human intracranial electrophysiological signals encoding rewards show a clear and ubiquitous context dependency, as has been observed in animals. To that end, we focused the inquiry on temporal context dependency and sought to gather evidence indicating whether or not the recent history of rewards influences the electrophysiological representation in reward-encoding areas of the human brain. Second, we sought to perform this search at the single electrode (contact) and within-subject levels, which might allow us to overcome some of the limitations faced by previous region-of-interest based human intracranial electrophysiology studies. Although averaging across subjects and electrode contacts has proven valuable in many previous studies, animal research suggests a limitation to this approach: while the averaged signal may encode a property like reward value, this representation may be non-uniformly distributed from micro-site to micro-site. We hypothesized that by analyzing data at the single contact level, many of the important features which have never before been examined in humans could be assessed as they have been in non-human animals. Third, by analyzing data at the single contact level and transforming all recording sites to a standard anatomical reference, we hypothesized that it might also be possible to assess the spatial distribution of reward network signals at a fine-grained level of analysis, as is common in animal research but has not yet been regularly undertaken in human studies.

Here we report the use of stereo electroencephalography (sEEG) to record neural activity in human epileptic patients ($n = 20$) performing an incentive compatible valuation task known to induce temporal context dependency at the behavioral level in humans[41]. Building on recent human intracranial work in decision-making[58,59], our

data show neurobiological evidence for temporal context-dependent value computations in humans. We observe this context dependency in a number of subregions of the OFC. High-gamma activity (80–150 Hz)—thought to aggregate heterogeneous neuronal activity near the recording site[60,61]—represents both the subjective value of the present reward under consideration and the subjective value of the reward offered on the previous trial. The same patterns of correlation also arise in the gamma band (30–80 Hz). Our single-contact analysis reveals that at the level of gamma and high-gamma band signals, statistically significant single contacts encode either subjective value or temporal context, with only a few contacts encoding both. In other brain areas we examined, the hippocampus and insula also carried these signals at the level of activity averaged across contacts and at the single contact level. Our single contact mapping data revealed that, as in monkey data[61], high-frequency activity in only about 30% of recording sites carry statistically significant subjective value signals, and these sites are found to be distributed throughout each of the fronto-cortical and subcortical areas we examined. As in monkeys, not all locations within an area encode subjective value, and the locations which are not apparently spatially clustered but rather appear distributed throughout a given subarea.

Context dependency is a ubiquitous feature of a wide array of cognitive functions, from perception, action, memory, judgment and inference, to decision-making. Our results paint a novel and detailed picture of how context-dependent computations might be implemented in human frontal activity. First, by showing subjective value-related electrophysiological signals in the human brain at both the population and single-contact level, our findings indicate that value and context are simultaneously but patchily represented in the OFC, insula, and hippocampus at the scale of sEEG. These findings are, it should also be noted, broadly compatible with at least some computational models for how context dependency arises in the subjective-value network[62]. Second, by performing state-space analysis and revealing the temporal trajectories of OFC population activity in the low-dimensional value-context space, our results indicate that context can be seen as a force that affects decision-related signals in a direction separable from the current item under consideration. This conceptualization may serve as a general principle of context-dependent computations underlying a wide array of cognitive functions.

## Results

In order to obtain behavioral measures of subjective value at the single-trial level, the subjects performed a version of the Becker-DeGroot-Marschak (BDM) auction task—a standard incentive-compatible paradigm used to elicit subjective value[63]. On each trial, the subjects saw an image of a snack food item presented on a computer screen and had to indicate the maximum amount they were willing to pay for the snack food item (Fig. 1a). When using the BDM method, maximum amount is a widely-validated measure of the subjective value for the food reward.

### Bidding behavior

We found several interesting features in the subjects' willingness-to-pay. First, across all subjects, the distribution of willingness-to-pay appeared to be positively skewed (Fig. 1b). About 23% of the trials across all subjects had zero bids. Second, the willingness-to-pay in a trial was significantly affected by the willingness-to-pay in the previous trial: the larger the subjects' bid in a trial, the higher she or he tended to bid in the next trial (Fig. 1c) even though the order of the different rewards presented across trials was determined randomly, indicating a temporal context dependency in bids. For each subject separately, we performed a linear regression analysis using current bid as data and bid from the previous trial as the regressor. To examine whether the regression coefficient is significantly different from 0, we performed a one-sample $t$ test on the regression coefficient ($\alpha = 0.05$, two-tailed). Across all subjects, the $t$ statistics ranged from −1.72 to 5.37 (Supplementary

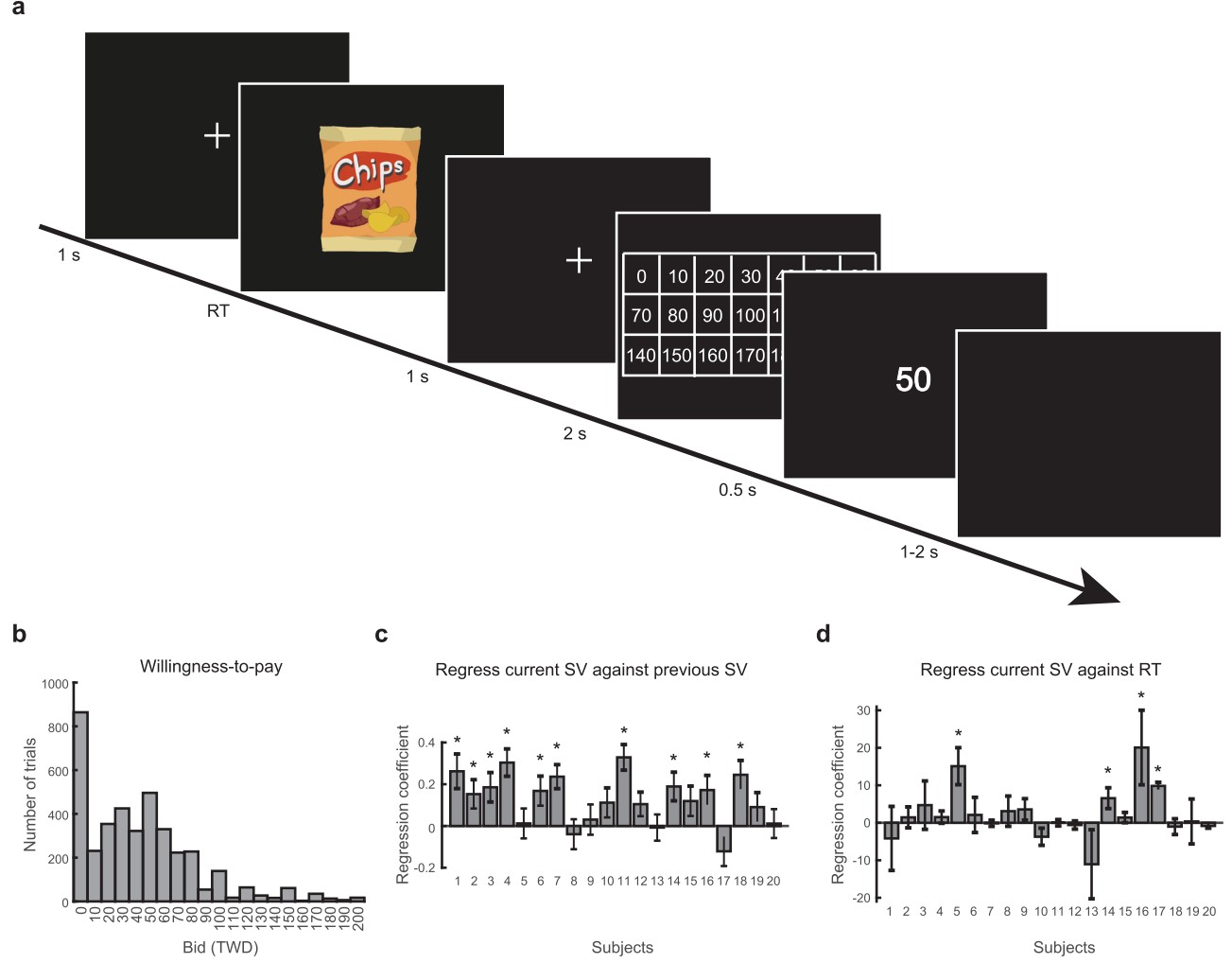

**Fig. 1 | Experimental design and behavioral results. a** Trial sequence of the Becker-DeGroot-Marschak (BDM) auction task. On each trial, the subjects faced a snack food item and had to indicate their willingness to pay for that item. Subjects first pressed the left button on the mouse to signal that they were ready to indicate their willingness-to-pay. A matrix that indicated possible prices, from 0 to 200 New Taiwan Dollars (TWD, 1 USD = 30 TWD), in 10-dollar increments, would then appear on the computer screen. The subjects' task was to use the mouse cursor to point to and click on the number closest to their maximum willingness-to-pay. **b** The distribution of willingness-to-pay from all subjects on all food items. **c** The impact of the bid offered by the subject on the previous trial on willingness-to-pay on the current trial. Note that sequential items were selected randomly and in an uncorrelated manner for presentation. For each subject, we regressed their willingness-to-pay—a measure of subjective value (SV)—in the current trial against the willingness-to-pay in the previous trial. Here we plot the regression coefficient of the willingness-to-pay in the previous trial. In the majority of subjects, willingness-to-pay in the current trial was positively correlated with the willingness-to-pay in the previous trial. **d** The relationship between the willingness-to-pay and response time (RT). For each subject, we regressed their willingness-to-pay against the response time in the trial. We plot the regression coefficient of the response time. In the majority of subjects, there was no relation between willingness-to-pay and response time. The * symbol indicates $p < 0.05$ (one-sample $t$ test, two-tailed). Error bars represent ±1 standard error of the mean. Source data are provided as a Source Data file.

Table 1 for reports on the $t$ statistics). At the single-subject level, 10 out of 20 subjects showed a significant effect of the previous subjective value (regression coefficients significantly different from 0 at $p < 0.05$ were marked with * in Fig. 1c). At the group level, the mean regression coefficient (across subjects) was significantly different from 0 (one-sample $t$ test, two tailed, $t = 4.77$, $p < 0.001$). Further analysis revealed that the bid in the trial presented two-trials back did not have a significant effect on the current bid (one-sample $t$ test, two-tailed, $t = 0.41$, $p = 0.341$; see Supplementary Fig. 1 and Supplementary Table 2). To further examine whether such temporal context dependency can be considered normal, we ran the same task on 35 healthy subjects from the normal population and found that most subjects (25 out of 35) showed the same temporal context dependency, with the current bid positively correlating with the bid on the previous trial (one-sample $t$ test, two-tailed, $t = 3.81$, $p < 0.001$; see Supplementary Fig. 2 and Supplementary Table 3; see also ref. 41). Third, we found no

relationship between response time (RT) (how long it took the subjects to place the bid) and the amount of their willingness-to-pay (Fig. 1d). For each subject, we performed a linear regression analysis using RT as data and willingness-to-pay as the regressor. A one-sample $t$ test ($\alpha = 0.05$, two-tailed) was performed on the regression coefficient of willingness-to-pay. Across all subjects, the $t$ statistics ranged from −1.63 to 10.20 (Supplementary Table 4 for reports on the $t$ statistics). At the single-subject level, 4 out of 20 subjects showed significant effect of RT (regression coefficient significantly different from 0 at $p < 0.05$ were marked with * in Fig. 1d). At the group level, the mean regression coefficient of RT was not significantly different from 0 (one sample $t$ test, two-tailed, $t = 1.60$, $p = 0.063$). We therefore concluded that there was no significant relation between RT and willingness-to-pay. The distribution of individual subjects' data on the willingness-to-pay and RT can be found in the Supplement (Supplementary Figs. 3, 4). Individual subjects' scatterplots on the willingness-to-pay of the current trial

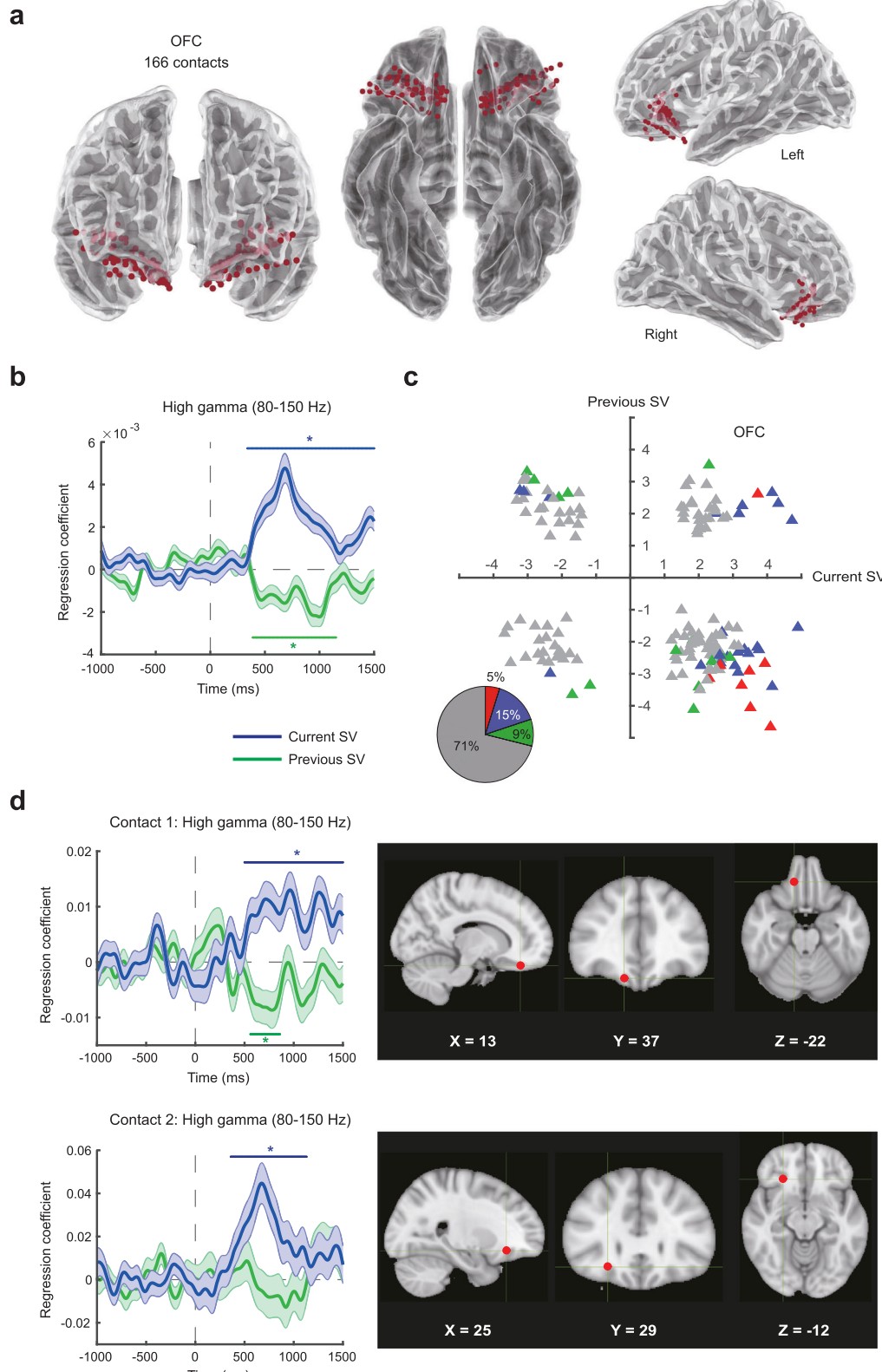

High-gamma activity in the orbitofrontal cortex represents past and present subjective value

In the OFC, we collected sEEG signals from a total of 166 electrode contacts in 20 subjects (Fig. 2a). The sEEG preprocessing pipeline can be found in the Supplement (Supplementary Fig. 7). After data pre-processing, we performed time-frequency analysis, separately for each trial, on the preprocessed local field potential (LFP) data in order to extract the timeseries data of oscillatory power associated with different frequency bands. The resulting timeseries data started from 1 s before the onset of the food stimulus to 1.5 s after the stimulus onset. The 1.5 s post-stimulus time window was approximately the average of

against that of the previous trial and on the willingness-to-pay against RT can also be found in the Supplement (Supplementary Figs. 5, 6).

**Fig. 2 | High-gamma activity (80–150 Hz) in the human OFC represents the subjective value of food rewards. a** Location of OFC electrode contacts mapped onto a standard reference brain (Montreal-Neurological-Institute, or MNI template). We used the toolbox in https://github.com/fahsuanlin/fhlin_toolbox to generate the brain images. We collected data from a total of 166 electrode contacts in the OFC from 20 subjects. **b** Subjective-value representations across OFC contacts. Here we plot the mean time course (averaged across all OFC contacts) of regression coefficients for the subjective value of the current trial (current SV, in blue) and the subjective value of the previous trial (previous SV, in green). **c** Subjective-value representations from individual OFC contacts. We plot the $t$ statistic of the current subjective value against that of the previous subjective value separately for each contact. Each data point represents a single contact. For each contact, we select the most significant time point according to the threshold-free-cluster-enhancement (TFCE) statistic, and plot the corresponding $t$ statistic, separately for the current subjective value and previous subjective value. Since the $t$ statistics come from the most significant time points, the data points in the graph are biased away from zero. Individual contacts that significantly represent the current subjective value, previous subjective value, or both are shown in blue, green, and red respectively. Individual contacts that neither represented the current nor the previous subjective value are shown in gray. The pie chart shows the proportions of contacts belonging to each of the categories described above. **d** Results from two example OFC contacts. Coordinates are in MNI space. We used FSL [Smith, S. M. et al. Advances in functional and structural MR image analysis and implementation as FSL. in NeuroImage 23 (2004)] to generate the brain images. Colored (blue or green) horizontal lines with the * symbol on top or beneath indicate the time points with $p < 0.05$ (familywise error corrected) using permutation test (one-tailed) with the threshold-free-cluster-enhancement (TFCE) statistic as the test statistic. Error bands represent ±1 standard error of the mean. Source data are provided as a Source Data file.

the median RT across all subjects (1.5011 s; Supplementary Fig. 8). We recognize that if we set a longer time window, we could capture the dynamics of valuation more comprehensively in trials with longer RTs. However, for shorter response-time trials, this would include time points after the subjects had already made a decision and entered into the response phase of the trial. Hence, choosing the time window at this length was an attempt to strike a balance between these opposing factors.

We subsequently performed General Linear Model (GLM) analyses on the power timeseries data with the subjective value of the current trial and the previous trial as two separate regressors. In particular, we focused on the high-gamma power (80–150 Hz) and referred to it as high-gamma activity to be consistent with the literature[61]. For all the neural GLM analyses, statistical significance was established based on a permutation test with threshold-free cluster enhancement (TFCE) as the test statistic[64,65]. The TFCE statistic is used to describe signals that exhibit some spatial or temporal continuity without having to arbitrarily set a threshold for defining clusters of signals. For a given regressor of interest, we computed the TFCE statistic based on the $z$-statistic timeseries of the corresponding regression coefficients and for each time point separately. To correct for multiple testing across time points, a familywise error rate at 0.05 was implemented based on comparing the TFCE statistic at each time point with the null distribution of the maximum (or minimum) TFCE statistic. See *Group-level permutation test (across electrode contacts)* in *Methods* for details. The time points that survived multiple-testing correction were marked with the * symbol (Figs. 2–8).

Across the OFC contacts, we found that high-gamma activity positively correlated with the current subjective value but negatively correlated with the previous subjective value (Fig. 2b; permutation test, one-tailed, $p < 0.05$, familywise error corrected for multiple testing across time points; see Table 1 for reports on the statistics). A version of these results in the form of the $t$ statistic of the regression coefficients can be seen in the Supplement (Supplementary Fig. 9).

The subjective-value representations were not only seen at the group level (across all OFC contacts), but also at the individual-contact level (Fig. 2c, d). We found that 29% of the OFC contacts (Fig. 2c) showed significant subjective-value representations (permutation test, one-tailed, $p < 0.05$, familywise error corrected for multiple testing across time points). See *Individual-level permutation test (for individual electrode contacts)* in *Methods* for details. The scatter plot of the $t$ statistics (Fig. 2c) according to the most extreme TFCE statistic–for each contact we plot the $t$ statistic of current subjective value that has the most extreme TFCE against the $t$ statistic of previous subjective value that has the most extreme TFCE statistic–revealed that the majority of significant OFC contacts cluster in the fourth quadrant, suggesting a positive correlation with the current subjective value and negative correlation with the previous subjective value. This result is consistent with the group-level results (Fig. 2b). Among the significant

OFC contacts, 52% significantly represented only the current subjective value, 31% of the contacts represented only the previous subjective value, and 17% represented both the current and previous subjective value. In other words, a majority of the significant contacts represented either the current or the previous subjective value, but not both. Data from two example contacts are also shown (Fig. 2d): One contact shows significant positive correlation with the current subjective value and negative correlation with the previous subjective value. The other contact exhibits only a positive correlation with the current subjective value.

The presence of the previous subjective-value representation is consistent with the view that the OFC is sensitive to the temporal context of experience[35]. It is also consistent with the results in monkey OFC[15,28] and with the view that the OFC implements a divisive-normalization algorithm to compute relative subjective value which we measure here using a simple linear regression[66].

## Robustness of subjective-value representations in the OFC

To examine the robustness of the subjective-value representations, we performed six additional analyses to rule out potential confounds. The significant results reported below were based on permutation test (one-tailed) using TFCE statistic as the test statistic ($p < 0.05$, familywise error corrected). The detailed statistical summary can be seen in Supplementary Tables 5 (for high-gamma activity) and 6 (for gamma activity).

First, we examined whether the results could have been driven by collinearity between the current and previous subjective value, since in most subjects the stated current subjective value positively correlated with the stated subjective value in the previous trial (Fig. 1c). To examine this possibility we carried out a regression analysis in two steps (GLM-2 in Fig. 3b). GLM-2 consisted of two separate regressions performed in two steps. In the first regression, we regressed high-gamma activity against only the current subjective value. We did not include the previous subjective value as regressor. Then, in the second regression, we used the residuals from the first regression as data and regressed them against only the previous subjective value. In this second regression, we did not include the current subjective value as regressor. We then plotted the regression coefficient of the current subjective value estimated from the first regression, and the regression coefficient of the previous subjective value estimated from the second regression (Fig. 3b). The results still indicated that the high-gamma activity in the OFC positively correlated with the current subjective value and negatively correlated with the previous subjective value, consistent with the original model (GLM-1 in Fig. 3a). We also performed this two-step regression analysis in the reversed direction (regress high-gamma activity against previous subjective value in the first regression and regress the residuals from the first regression against the current subjective value in the second regression) and the results were identical (see Supplementary Fig. 10).

**Table 1 | Summary of statistical analysis examining the effects of current and previous subjective value on high-gamma activity**

| ROI | Regressor | Cluster size | Start time (ms) | End time (ms) | Maximum or minimum TFCE | p value | Peak time (ms) |
|---|---|---|---|---|---|---|---|
| OFC | Current SV | 117 | 340 | 1500 | 17,237,135 | <0.0001 | 710 |
| OFC | Previous SV | 77 | 390 | 1150 | −1,477,123 | 0.0003 | 1010 |
| Medial OFC | Current SV | 65 | 400 | 1040 | 857,411 | 0.0039 | 690 |
| Medial OFC | Previous SV | 78 | 640 | 1410 | −953,291 | 0.0003 | 1020 |
| Central OFC | Current SV | 118 | 330 | 1500 | 11,840,031 | <0.0001 | 850 |
| Central OFC | Previous SV | 78 | 370 | 1140 | −1,090,378 | 0.0013 | 890 |
| Lateral OFC | Current SV | 20 | 570 | 760 | 398,460 | 0.0357 | 660 |
| Lateral OFC | Current SV | 16 | 1000 | 1150 | 403,743 | 0.0353 | 1110 |
| Anterior OFC | Current SV | 136 | 150 | 1500 | 5,448,448 | <0.0001 | 1010 |
| Anterior OFC | Previous SV | 16 | −670 | −520 | −371,424 | 0.0442 | −570 |
| Anterior OFC | Previous SV | 61 | −320 | 280 | 699,254 | 0.0177 | −220 |
| Anterior OFC | Previous SV | 108 | 380 | 1450 | −921,352 | 0.0086 | 660 |
| Posterior OFC | Current SV | 75 | 360 | 1100 | 6,125,627 | <0.0001 | 680 |
| Posterior OFC | Current SV | 20 | 1310 | 1500 | 373,035 | 0.0298 | 1480 |
| Posterior OFC | Previous SV | 80 | 410 | 1200 | −2,315,647 | 0.0001 | 990 |
| Amygdala | Current SV | 15 | 560 | 700 | 218,651 | 0.0357 | 590 |
| Hippocampus | Current SV | 9 | −810 | −730 | −289,468 | 0.0455 | −770 |
| Hippocampus | Current SV | 13 | −690 | −570 | −368,070 | 0.0308 | −630 |
| Hippocampus | Current SV | 29 | 20 | 300 | −508,621 | 0.0177 | 200 |
| Hippocampus | Current SV | 18 | 360 | 530 | −332,990 | 0.0346 | 450 |
| Hippocampus | Previous SV | 63 | 290 | 910 | −1,943,490 | 0.0003 | 760 |
| Insula | Current SV | 36 | 450 | 800 | 439,686 | 0.0155 | 630 |
| Insula | Previous SV | 23 | 1190 | 1410 | 238,403 | 0.0277 | 1230 |
| ACC & MCC | Previous SV | 11 | 940 | 1040 | −221,955 | 0.0359 | 980 |
| PCC | Previous SV | 19 | 710 | 890 | −207,341 | 0.0314 | 850 |

Statistical significance was established based on a permutation test with threshold-free cluster enhancement (TFCE) as the test statistic. Details of the statistical procedure can be found in *Methods*. Here cluster size refers to the size of temporal cluster, i.e., number of consecutive time points whose p value are less than 0.05 after familywise error correction for multiple testing across time points. Since TFCE was calculated based on the z statistic, TFCE would be positive for positive effects, and negative for negative effects. The maximum TFCE indicates the strongest positive effect within the temporal cluster, while the minimum TFCE indicates strongest negative effect. The p value was estimated based on the null distribution of the maximum TFCE statistic (for positive effects) or the null distribution of the minimum TFCE statistic (for negative effects) through permutations. The start and end time indicate the start and end time of the temporal cluster. Time at 0 indicates the onset of food stimulus presentation. The peak time corresponds to the time point with either the maximum or minimum TFCE within the temporal cluster. Source data are provided as a Source Data file.

Second, we examined whether subjective-value representations can be affected by the zero-bid trials, as these trials represented 23% of the total trials gathered across subjects (Fig. 1b). In GLM-3 we therefore excluded the zero-bid trials and only included the non-zero bid trials in the analysis (Fig. 3c). The results were again consistent with the original model. Third, we examined whether RT might somehow interact with subjective-value in a way that altered the results found in GLM-1. Therefore, in GLM-4, in addition to the current and previous subjective value as regressors, we added the subjects' current-trial RT as a regressor to the model. Again, the results (Fig. 3d) were consistent with the original model (GLM-1 shown in Fig. 3a). Due to the difference in scale between the SVs and RT, the vertical axis in Fig. 3d on the left represents the regression coefficient of the current (blue) and previous (green) SV, whereas the vertical axis on the right represents the regression coefficient of RT. Fourth, we extended the original model by adding the subjective value obtained two trials previously as a regressor so as to examine whether the results would be consistent with the original results and to also examine the impact of the two-trials back bid on the OFC activity (Fig. 3e). The results were consistent with the original findings. We found no significant impact of the subjective value in the two-trial back on the OFC activity. While this almost certainly reflects a power issue, we are unable to conclude from this result whether or not the impact of previous trials extends back beyond one previous trial. Fifth, we examined whether the negative correlation with the previous subjective value could arise from signal autocorrelation, the deterministic correlation in spectral power between two successive trials. To address this issue we modified GLM-1 by adding the power on the previous trial as a regressor in addition to

the current and previous subjective value regressors. That is, for each time point separately, the spectral power of the same time point from the previous trial was used as the regressor. We found that the negative correlation with previous subjective value remained (Fig. 3f). Due to the difference in scale between the SVs and spectral power from the previous trial, the vertical axis on the left in Fig. 3f represents the regression coefficient of the current (blue) and previous (green) SV, whereas the vertical axis on the right represents the regression coefficient of power from the previous trial. Sixth, we examined whether the results on the previous subjective value could be affected by the length of the inter-trial interval (ITI) that we randomly varied across trials (1 s, 1.5 s, and 2 s). If signal autocorrelation did have an effect on the previous-value findings, we expect to observe weaker results on previous subjective value with shorter ITIs. We implemented GLM-1 separately for each possible ITI and found that the length of ITI did not affect results on the previous subjective value (Fig. 3g). The negative correlation with previous subjective value was significant across different ITIs. Further analyses on issues related to signal autocorrelation can be seen in the Supplement (Supplementary Figs. 11, 12). Finally, we noticed that the patterns of subjective-value representations were similar between the high-gamma (80-150 Hz) activity (Fig. 3) and the gamma (30–80 Hz) activity (Supplementary Fig. 13) across these different regressions. The t-statistic version of these results (Fig. 3 and Supplementary Fig. 13) can be seen in the Supplement (Supplementary Fig. 14). This suggested that activity in a broad frequency range (from 30–150 Hz) represented the subjective value of food rewards in a similar fashion. Together, these robustness checks support the conclusions that OFC electrophysiological activity in humans encodes

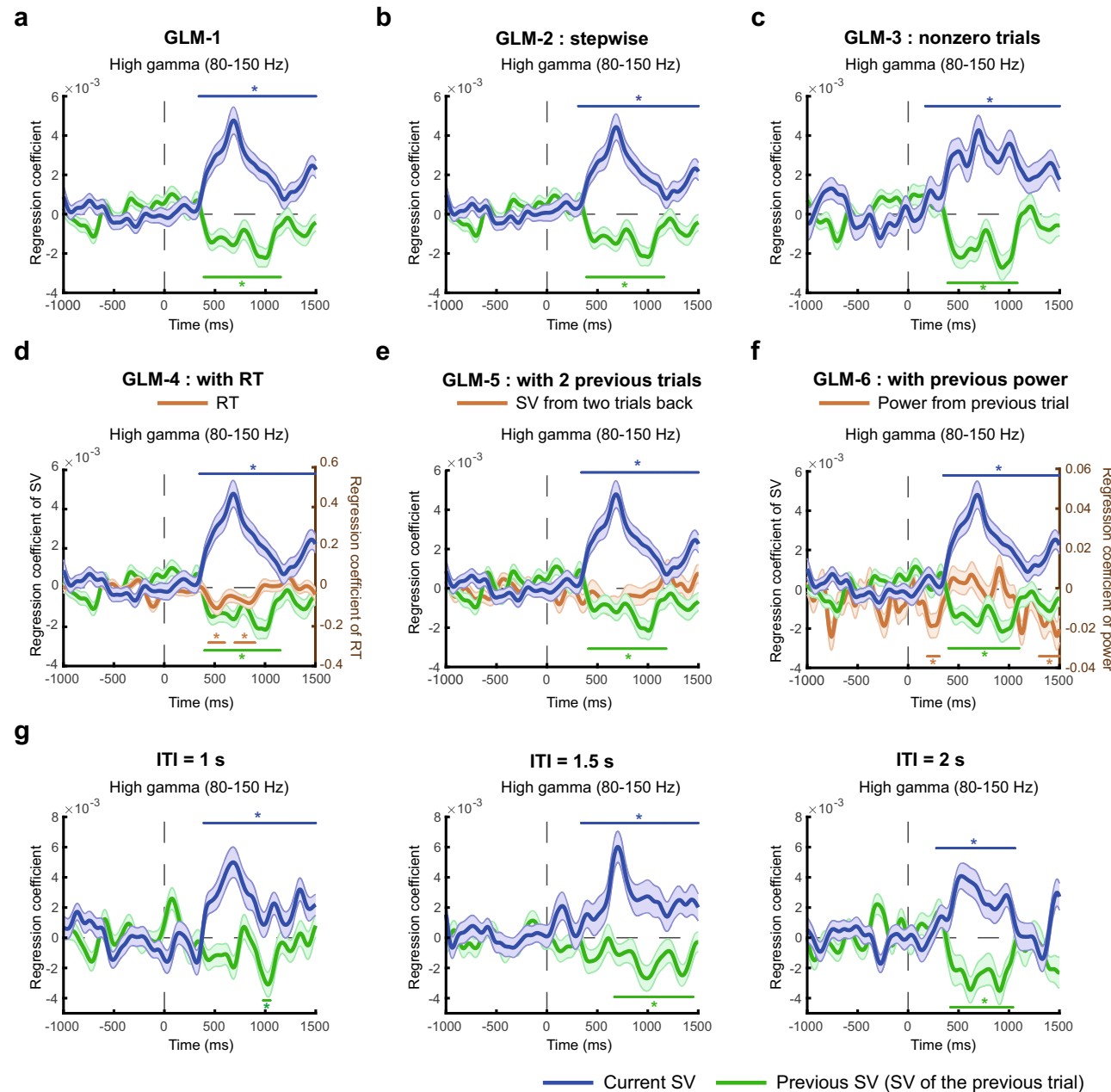

**Fig. 3 | Testing the robustness of subjective-value representations in the OFC.** We performed seven different General-Linear-Modeling (GLM) analyses to examine the robustness of subjective-value representations in the OFC. The GLMs were implemented for the high-gamma activity (80–150 Hz). **a** GLM-1. This is the original model where brain activity was regressed against the current and the previous subjective value. The high-gamma graph was identical to that shown in Fig. 2b. **b** GLM-2. This analysis was performed in two steps. In the first step, we regressed brain activity against only the current subjective value. In the second step, we used the residuals from the first step and regressed them against only the previous subjective value. **c** GLM-3. The model was identical to GLM-1 except that we only included trials where the subjects' willingness-to-pay were not zero in the analysis. **d** GLM-4. The model was identical to GLM-1 except that the subjects' response time

(RT) in the current trial was added as a regressor to the model. **e** GLM-5. The model was identical to GLM-1 except that we added the subjective value of the option encountered two-trials back as a regressor. **f** GLM-6. The model was identical to GLM-1 except that we included the spectral power from the previous trial as a regressor. **g** Evaluating GLM-1 at different inter-trial intervals (ITIs). We sorted trials according to the preceding ITI (1 s, 1.5 s, or 2 s) and estimated GLM-1 separately for each possible ITI. Error bands represent ±1 standard error of the mean. Colored horizontal lines with the * symbol on top or beneath indicate the time points with $p < 0.05$ (familywise error corrected) using permutation test (one-tailed) with the threshold-free-cluster-enhancement (TFCE) statistic as the test statistic. Source data are provided as a Source Data file.

both the subjective-value of the currently offered option and the subjective-value of at least the most recently considered option.

**Subjective-value representations in different OFC subregions**
We next asked whether patterns of subjective-value representations differed between the subregions of the human OFC. We examined the medial-to-lateral OFC (Fig. 4) and the anterior-to-posterior OFC (Fig. 5).

For medial-to-lateral OFC, three major subregions—the medial (area 14), central (areas 11 and 13), and lateral (area 47/12) OFC—were identified by consulting the automated anatomical atlas 2 (AAL2)[67]. In all three regions, high-gamma activity significantly correlated with the current subjective value (Fig. 4 in blue; permutation test, one-tailed, $p < 0.05$, familywise error corrected for multiple testing across time points; see Table 1 for reports on the statistics). By contrast, not all

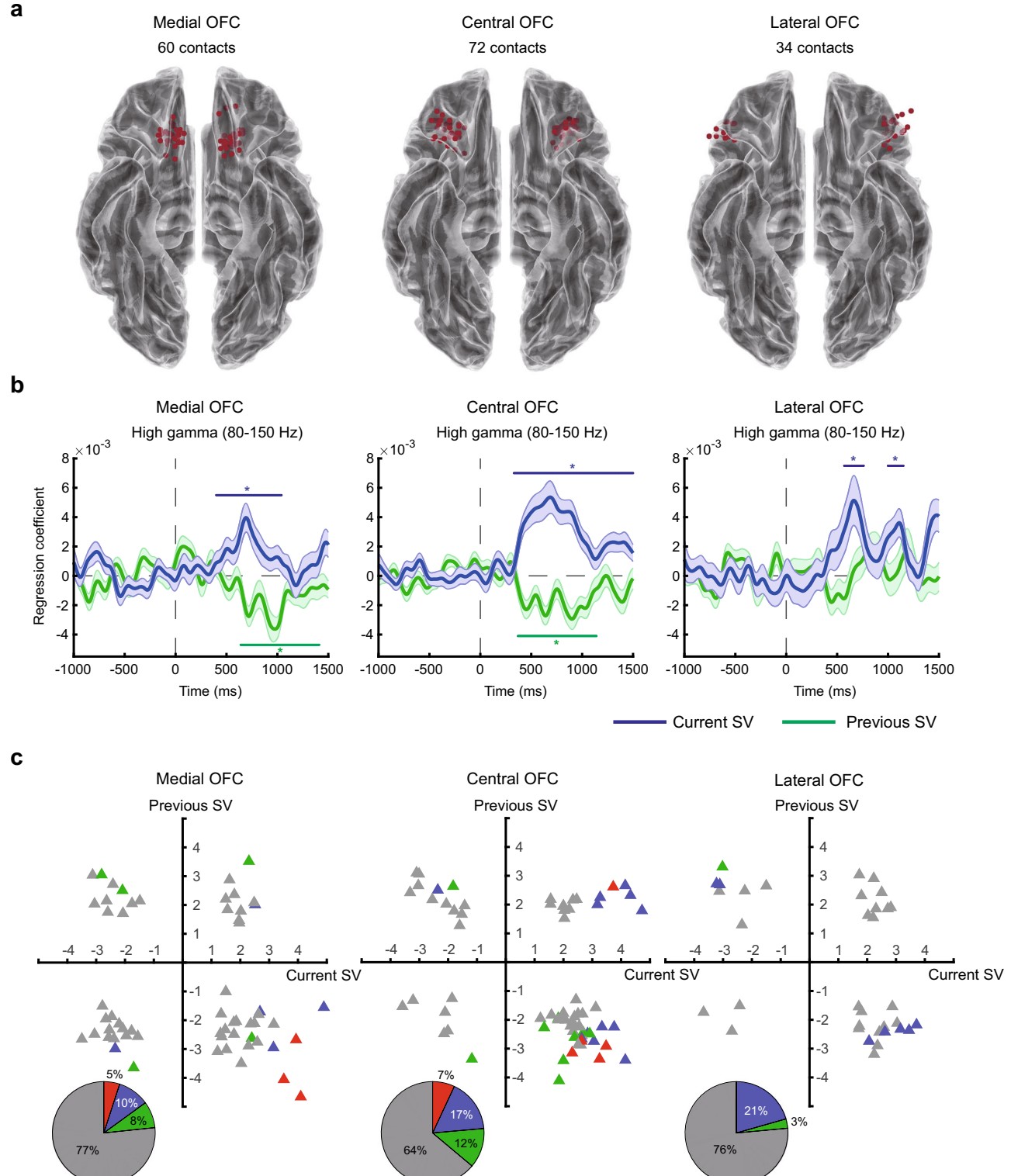

**Fig. 4 | Subjective-value representations in the medial, central, and lateral OFC. a** Electrode contacts in the medial (left), central (middle), and lateral OFC (right). **b** High-gamma activity. Average time course of regression coefficients for the current subjective value (in blue) and previous subjective value (in green) in the medial, central, and lateral OFC. Colored (blue or green) horizontal lines with the * symbol on top or beneath indicate the time points with *p* < 0.05 (familywise error corrected) using permutation test (one-tailed) with the threshold-free-cluster-enhancement (TFCE) statistic as the test statistic. **c** Subjective-value representations in individual OFC contacts. Error bands represent ±1 standard error of the mean. Source data are provided as a Source Data file. We used the toolbox in https://github.com/fahsuanlin/fhlin_toolbox to generate the brain images.

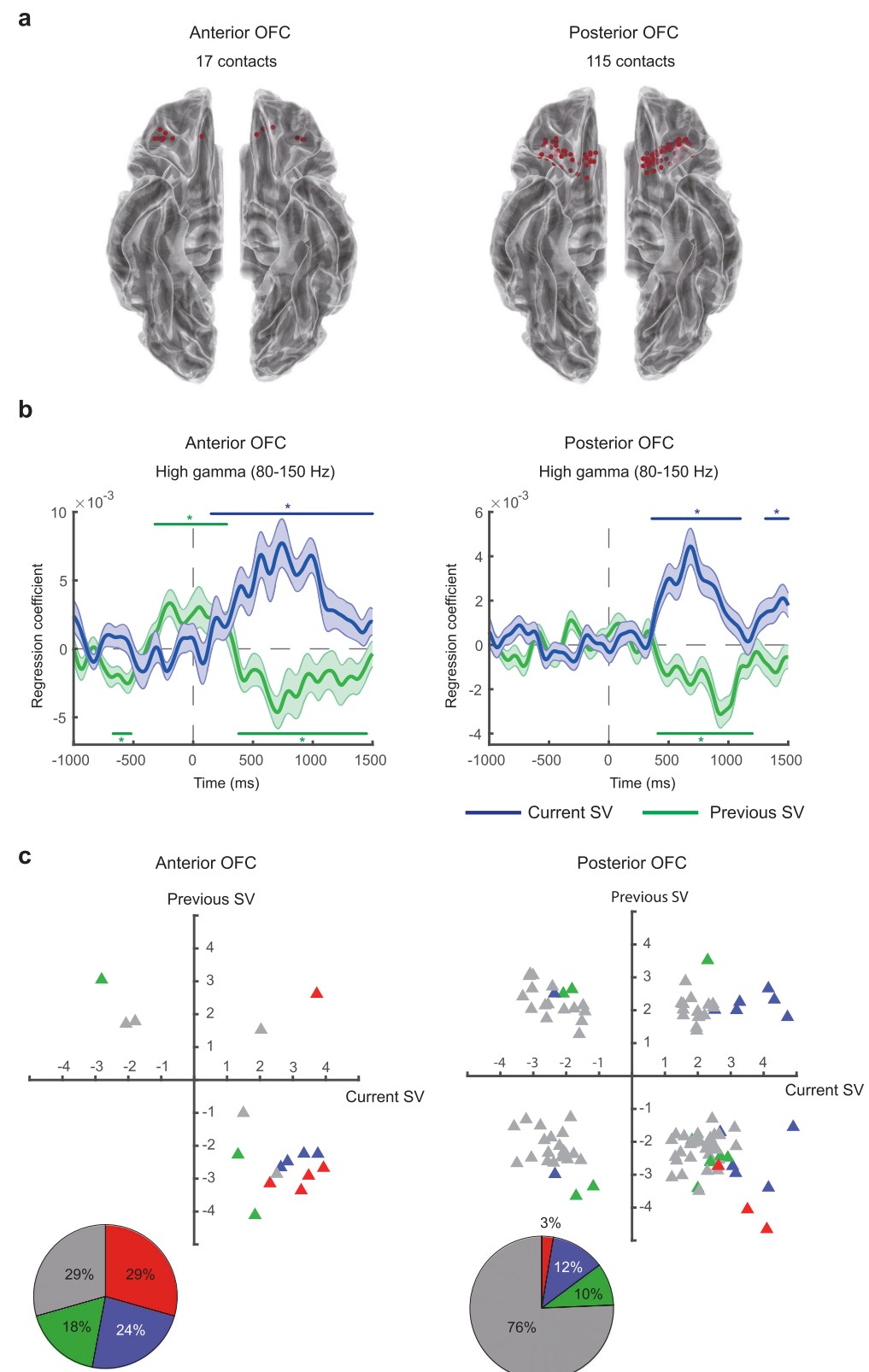

**Fig. 5 | Subjective-value representations in the anterior and posterior OFC.**
**a** Electrode contacts in the anterior (left) and posterior OFC (right). **b** High-gamma activity. Average time course of regression coefficients for the current subjective value (in blue) and previous subjective value (in green) in the anterior and posterior OFC. Colored (blue or green) horizontal lines with the * symbol on top or beneath indicate the time points with $p < 0.05$ (familywise error corrected) using permutation test (one-tailed) with the threshold-free-cluster-enhancement (TFCE) statistic as the test statistic. **c** Subjective-value representations in individual OFC contacts. Error bands represent ±1 standard error of the mean. Source data are provided as a Source Data file. We used the toolbox in https://github.com/fahsuanlin/fhlin_toolbox to generate the brain images.

regions showed significant representation of the subjective value encountered in the previous trial: both the medial and central OFC significantly and negatively correlated with the previous subjective value (permutation test, one-tailed, $p < 0.05$, familywise error corrected for multiple testing across time points; see Table 1 for reports on the statistics), but not the lateral OFC (Fig. 4 in green). At the individual-contact level, the majority of contacts that significantly represented the current subjective value (blue triangles) showed positive correlation with the current subjective value (permutation test, one-tailed, $p < 0.05$, familywise error corrected for multiple testing across time points; see Table 1 for reports on the statistics). The results on the previous subjective value, at the individual-contact level, were less consistent across the three regions. In the central OFC, significant previous-value contacts tended to show a negative correlation with previous subjective values. In the medial and lateral OFC, this tendency was less obvious.

For the anterior-to-posterior axis, we focused on electrode contacts in the central and medial OFC because of previous results from the monkeys[61]. For each contact in the central and medial OFC, we simply labeled it as either anterior or posterior OFC. We consulted AAL2[67] and elected to use $y = 35$ in the MNI coordinates as the anterior-posterior boundary. As a result, we identified 17 contacts in the anterior OFC and 115 contacts in the posterior OFC (Fig. 5a). We found that, similar to the overall results, both the anterior and posterior OFC positively correlated with the current subjective value, but negatively correlated with the previous subjective value (Fig. 5b; permutation test, one-tailed, $p < 0.05$, familywise error corrected for multiple testing across time points; see Table 1 for reports on the statistics). At the single-contact level (Fig. 5c), the posterior OFC results were similar to the overall results shown in Fig. 2c (permutation test, one-tailed, $p < 0.05$, familywise error corrected for multiple testing across time points; see Table 1 for reports on the statistics), while the anterior OFC had relatively more statistically significant contacts. This, however, could simply be due to the fact that there were fewer contacts in the anterior OFC.

## Cross-frequency representations of subjective value

Surprisingly, in the OFC, we not only found significant subjective-value representations in the high-gamma and gamma activity, but also in the activity of lower frequencies (Fig. 6) (permutation test, one-tailed, $p < 0.05$, familywise error corrected for multiple testing across time points; see Supplementary Tables 7, 8 for reports on the statistics). The two-dimensional heatmap (Fig. 6a) plots the $z$ statistic of the regression coefficients (across all electrode contacts across all subjects) in the time-frequency space for the current subjective value (left graph in Fig. 6a) and for the previous subjective value (right graph in Fig. 6a). See *Group-level permutation test (across contacts) in the time-frequency space* in *Methods* for details. The colors in the maps reveal the encoding directions of subjective value—orange for positive correlation with the subjective value, blue for the negative correlation, and green for non-significant results. It is evident that, after stimulus onset (indicated by 0 on the horizontal axis), activity in the gamma and high-gamma band positively correlated with the current subjective value (the orange clusters in the left graph, Fig. 6a) but negatively correlated with the previous subjective value (the blue clusters in the right graph, Fig. 6a) (permutation test, one-tailed, $p < 0.05$, familywise error corrected for multiple testing across time points; see Supplementary Table 8 for reports on the statistics). Interestingly, the encoding patterns of subjective value were reversed in the low frequency bands. Activity in the beta (13–30 Hz), alpha (8–12 Hz), and theta (4–7 Hz) bands negatively correlated with the current subjective value (blue clusters in the left graph, Fig. 6a), but positively correlated with the previous subjective value (orange clusters in the right graph, Fig. 6a) (permutation test, one-tailed, $p < 0.05$, familywise error corrected for multiple testing across time points; see Supplementary Table 8 for

reports on the statistics). These results are further summarized in the group-level time series plots of the regression coefficients (Fig. 6b, Supplementary Table 7). At the individual-contact level, scatter plots of the $t$ statistic according to the most extreme TFCE statistic are plotted in Fig. 6c. Noticeably, the low-frequency activity significantly represented a bias on the current subjective value before stimulus onset—before information about the current food item was revealed. We found that in the alpha band, these results were associated with two behavioral patterns: the variability of the bids and the correlation between the subjects' stated current subjective value and the subjects' stated previous subjective value. As might be expected, we found that, in part, such pre-stimulus representations were driven by the subjects who showed less variability in their bids (Supplementary Fig. 15) and whose bids were more affected by the bid in the previous trial (Supplementary Fig. 16). In other words, the more the subjects relied on the previous bid, the greater the likelihood of significant pre-stimulus representations (a form of bias).

## Subjective-value representations in other brain regions

We also examined subjective-value representations in several other subcortical (Fig. 7) and cortical regions (Fig. 8) in which our participants had had contacts placed. The subcortical regions included the amygdala, hippocampus, and striatum. At the group-level (middle graphs in Fig. 7), all of these regions except the striatum showed significant subjective-value representations (permutation test, one-tailed, $p < 0.05$, familywise error corrected for multiple testing across time points; see Table 1 for reports on the statistics). Both the amygdala and the hippocampus significantly represented the current subjective value, while the hippocampus also represented the previous subjective value. In the hippocampus, at the individual-contact level, the majority of the significant contacts seemed to cluster in the fourth quadrant, suggesting a positive correlation with the current subjective value and a negative correlation with the previous subjective value—a result consistent with our findings in the OFC. In the amygdala, 90% of the contacts were non-significant, even though the group-level results indicate significant current-value representation in the area. To summarize, even though all three subcortical regions showed evidence for subjective-value representations, only the hippocampus showed significant representations for both the current and previous subjective value, at the group level and at the level of individual contacts.

In our analysis of recordings from cortical regions other than the OFC—including the insula, the ACC and midcingulate cortex (MCC), posterior cingulate cortex (PCC), and the intraparietal sulcus (IPS)—we found that, at the group level, all but the IPS significantly represented the subjective value (Fig. 8) (permutation test, one-tailed, $p < 0.05$, familywise error corrected for multiple testing across time points; see Table 1 for reports on the statistics). The insula significantly represented both the current and the previous subjective value, while the ACC-MCC, and PCC represented the previous subjective value. At the individual-contact level, about 85% of contacts in the ACC-MCC, PCC, and IPS were non-significant, while about 30% of the contacts in the insula were significant in either representing the current or the previous subjective value (permutation test, one-tailed, $p < 0.05$, familywise error corrected for multiple testing across time points; see Table 1 for reports on the statistics). The proportion of significant contacts in the insula (27%, Fig. 8a) was identical to the OFC (29%, Fig. 2c) and hippocampus (28%, Fig. 7b). In summary, while there was evidence for subjective-value representations in these four cortical regions, only the insula showed prominent representations for both the current and previous subjective value, as these representations were observed at the group level and at the level of individual contacts in our valuation task. The cross-frequency representations of subjective value associated with these brain regions, like the ones shown in Fig. 6a for the OFC, can be seen in the Supplement (Supplementary Figs. 17, 18).

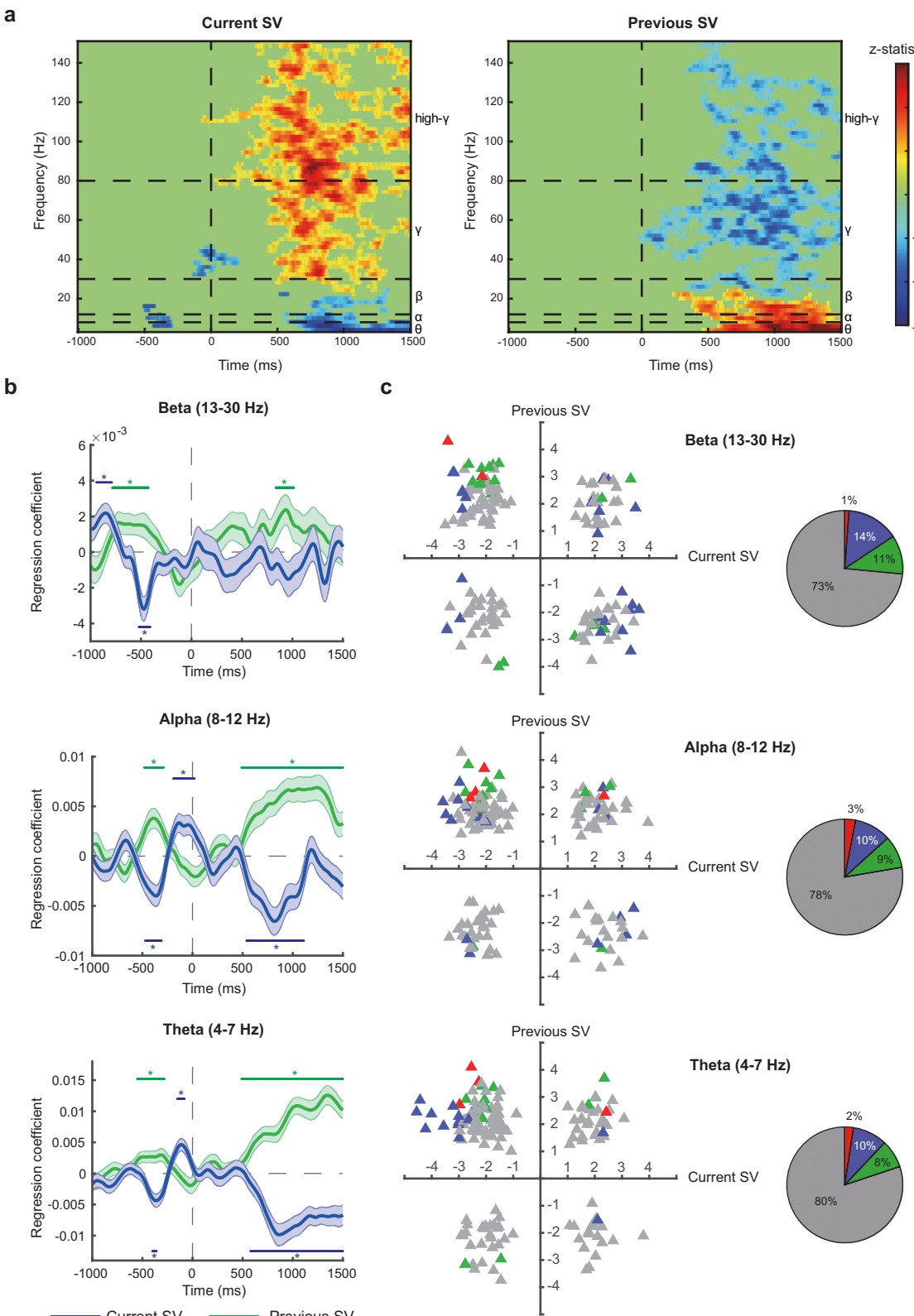

**Fig. 6 | Cross-frequency representations of subjective value. a** Time-frequency representations for subjective value. The heatmap plots the *z* statistic of the regression coefficient for the current subjective value (left graph) and previous subjective value (right graph). Significant clusters in these two-dimensional maps were identified by contiguous points in the time-frequency space that survived multiple testing with a familywise error rate of 0.05 according to the threshold-free-cluster-enhancement (TFCE) statistic (permutation test, one-tailed; Orange: significant positive correlation; blue: significant negative correlation; green: non-significant results). **b, c** Results from activity in the beta, theta, and alpha bands.

**b** Average time course of regression coefficients for the current subjective value (blue) and previous subjective (green) in the beta (13–30 Hz, top graph), alpha (8–12 Hz, middle graph), and theta (4–7 Hz, bottom graph) activity. **c** Subjective-value representations on individual OFC contacts in the beta (top graph), alpha (middle graph), and theta activity (bottom graph). Colored (blue or green) horizontal lines with the * on top or beneath indicate the time points with $p < 0.05$ (familywise error corrected) using permutation test (one-tailed) with the threshold-free-cluster-enhancement (TFCE) statistic as the test statistic. Error bands represent ±1 standard error of the mean. Source data are provided as a Source Data file.

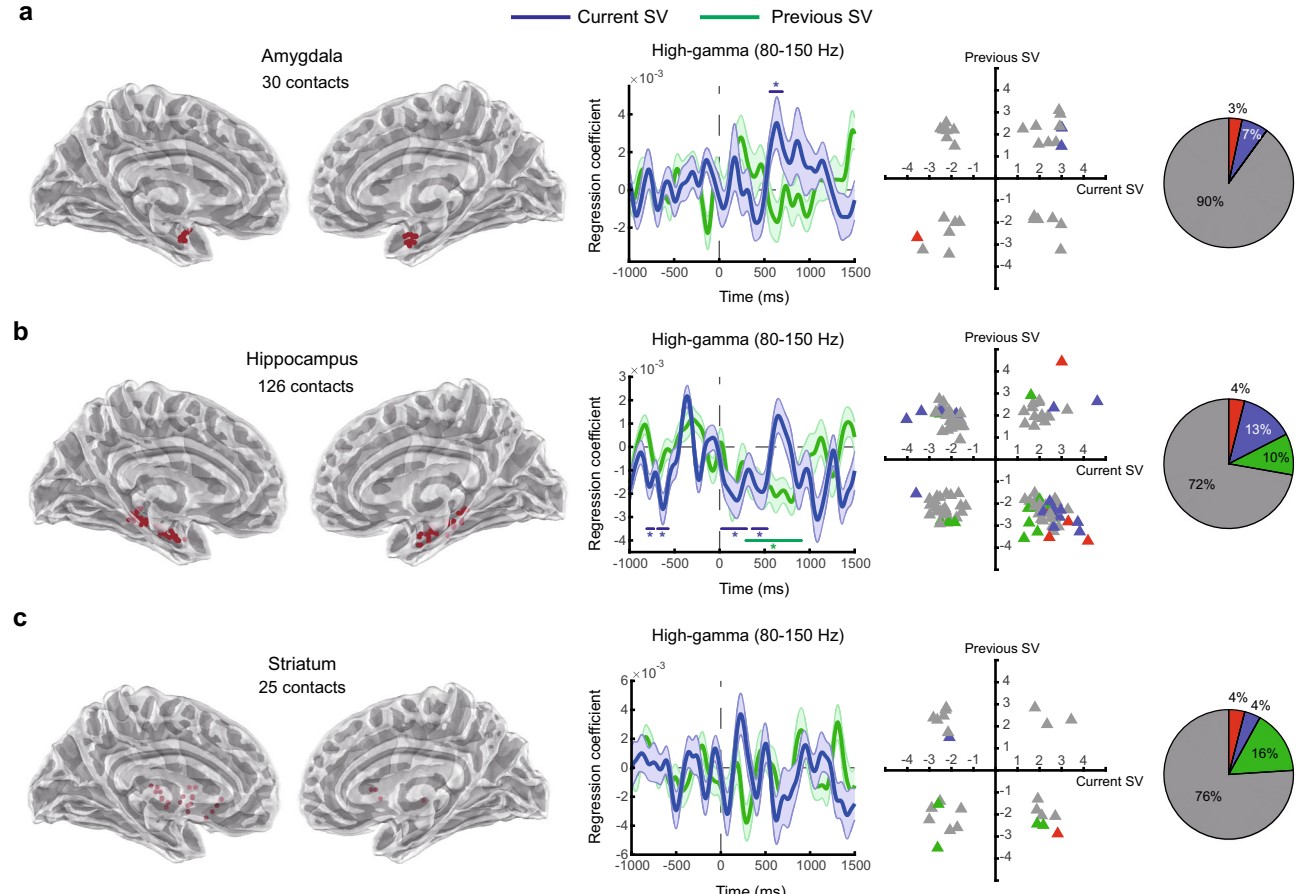

**Fig. 7 | Subcortical representations for subjective value.** High-gamma activity in the amygdala, hippocampus, and striatum. Conventions are the same as described in Fig. 2. **a** Amygdala. **b** Hippocampus. **c** Striatum. Colored (blue or green) horizontal lines with the * symbol on top or beneath indicate the time points with *p* < 0.05 (familywise error corrected) using permutation test (one-tailed) with the threshold-free-cluster-enhancement (TFCE) statistic as the test statistic. Error bands represent ±1 standard error of the mean. Source data and statistics are provided as a Source Data file. We used the toolbox in https://github.com/fahsuanlin/fhlin_toolbox to generate the brain images.

## OFC activity trajectories in a low-dimensional value-context space

To further understand how the OFC as a whole dynamically represents subjective value and context, we performed two final complementary analyses, one based on a principal component analysis (PCA) and the other based on a regression subspace analysis[68,69]. In these analyses, we approached the OFC activity from different electrode contacts as a single population consisting of high-dimensional data (each contact serves as a dimension). Through dimensionality reduction, these analyses allowed us to characterize the dynamics of the high-dimensional data as temporal trajectories in some low-dimensional space. An important question we asked is whether and how the axes forming the low-dimensional space relate to our task variables, namely the subjective value and temporal context. These analyses provided two major compliments to the GLM analyses. First, they serve to validate whether subjective value and temporal context dominated the electrophysiological signals in OFC as a population, as is implied by our regression analyses and by the structure of the task we employed. Second, they provided alternative visualizations for how subjective value and temporal context impacted OFC population activity.

For each electrode contact, we first computed the mean high-gamma activity timeseries (across all trials) and subtracted it from each individual trials' timeseries. Second, we sorted trials according to the subjective value of the current trial (high or low, median spilt) and the subjective value of the previous trial (high or low, median split). The medians we used were subject-specific. This resulted in a 2 (current subjective value magnitude) × 2 (previous subjective value magnitude) design and a total of four conditions. Third, we gathered the activity timeseries across all OFC contacts according to condition (Fig. 9a). Each condition is represented by a two-dimensional matrix where each row represents the timeseries−from 1 s before stimulus onset to 1.5 s after stimulus onset−of a single OFC electrode contact. Fourth, we stacked up the four two-dimensional matrices (one for each condition) and performed a single PCA across all conditions. The PCA allowed us to identify the dimensions in the neural state space that captured the most variance in the OFC high-gamma activity. Here the neural state space is a $N_{contacts}$-dimensional space where $N_{contacts}$ is the number of electrode contacts in the OFC ($N_{contacts}$ = 166). This analysis approach combines two advantages in our dataset − the idiosyncratic preferences of different individuals and a large number of electrode contacts across participants. In other words, by using the individual-specific medians to categorize trials into different conditions, we preserved the individual-specific preference information in this population-level, across-subjects analysis.

We then projected the activity timeseries from each of the four conditions of trials (current subjective value magnitude: high and low × previous subjective value magnitude: high and low) separately onto the first principal component (PC-1) and the second principal component (PC-2) and plotted a temporal trajectory of the population activity during each of these four conditions of trials (Fig. 9b). Time is indexed by the darkness of the colors, with the starting time point (1 s before stimulus onset) being the darkest and the end time point the

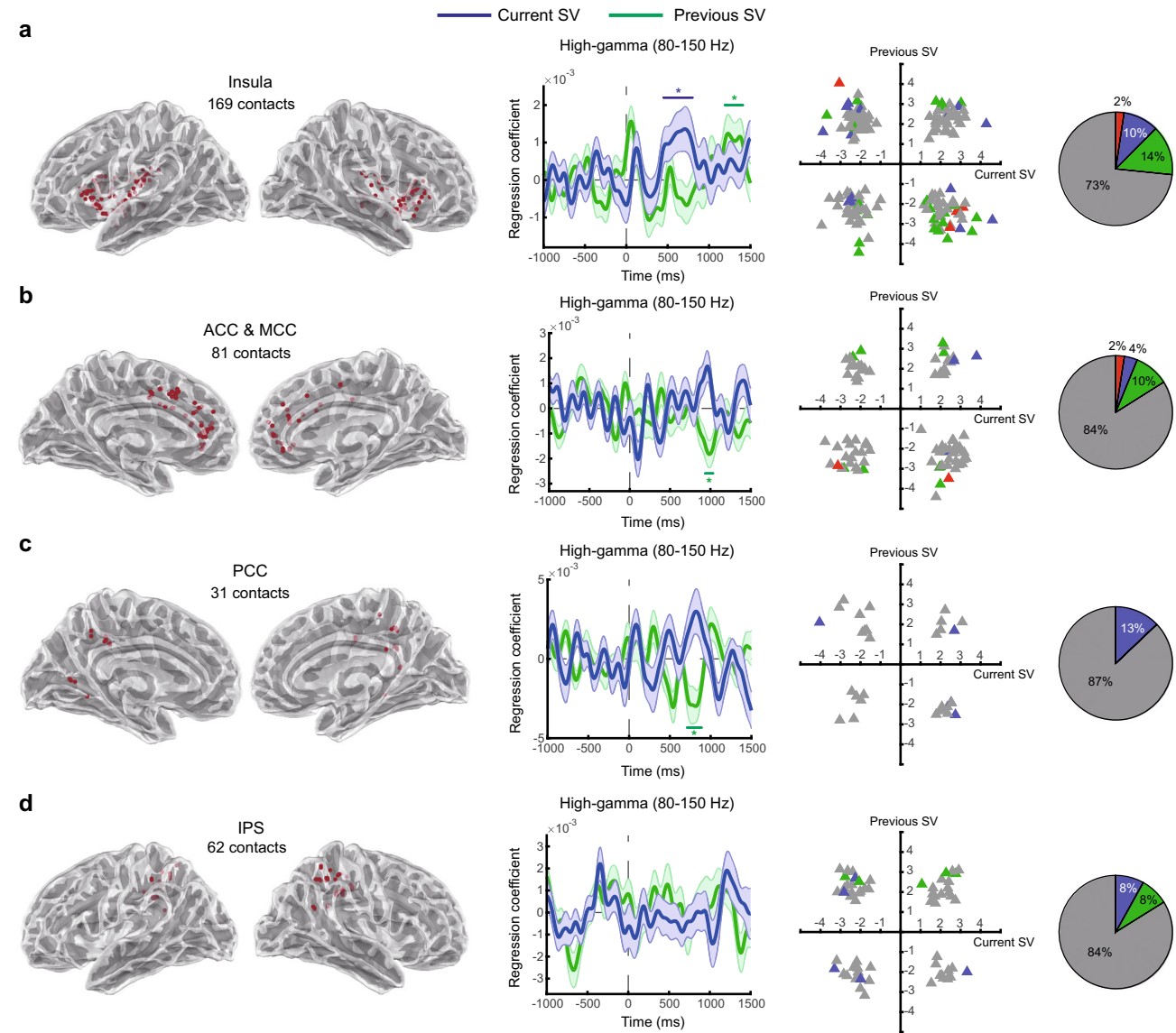

**Fig. 8 | Cortical representations for subjective value.** High-gamma activity in the insula, cingulate cortex, and intraparietal sulcus. Conventions are the same as described in Fig. 2. **a** Insula. **b** Anterior cingulate and midcingulate cortex (ACC and MCC). **c** Posterior cingulate cortex (PCC). **d** Intraparietal sulcus (IPS). Colored (blue or green) horizontal lines on top or beneath indicate the time points with $p < 0.05$ (familywise error corrected) using permutation test (one-tailed) with the threshold-free-cluster-enhancement (TFCE) statistic as the test statistic. Error bands represent ±1 standard error of the mean. Source data and statistics are provided as a Source Data file. We used the toolbox in https://github.com/fahsuanlin/fhlin_toolbox to generate the brain images.

lightest. Each data point in the trajectories represents a 50-ms time point. We found that each of the four conditions had very unique trajectories, suggesting that the top two principal components—the factors that accounted for the most variance in the OFC high-gamma activity—strongly and orthogonally capture information about current and previous-trial subjective value. The four activity trajectories start from a common origin (corresponding to 1 s before stimulus onset) and then diverge in four different cardinal directions. These directions revealed that the first principal component appears to capture the temporal context—subjective value observed in the previous trial (Fig. 9c), separating high previous subjective value (yellow and cyan) from low previous subjective value (purple and magenta). The degree of separation appears to depend on the current subjective value, with stronger separation between high and low previous subjective values when the current subjective value was high (cyan and magenta). The onset of this stronger separation emerged early, right after the stimulus onset (0 s mark). The second principal component appears to capture information in the population about the magnitude of the

subjective value observed in the current trial (Fig. 9d), separating high current subjective value (cyan and magenta) from low current subjective value (yellow and purple). The emergence of this separation also appeared to be early, ~200 ms after stimulus onset. These patterns were consistently observed under different data-smoothing parameters (Supplementary Fig. 19). This unbiased PCA analysis thus seems to support the notion that, at least when human subjects are performing a valuation task, much of the variability in activity observed in the OFC encodes the current value and context, with context information seeming to have a greater overall impact on the data variance.

Interestingly, the regression subspace analysis adapted from ref. 68 revealed a similar pattern of trajectories (Fig. 9e–g) as those observed in the PCA (Fig. 9b–d). The regression subspace analysis aims to reveal OFC population activity in a value-context low-dimensional subspace that captures the across-trial variance due to the subjective value of the current trial and the subjective value of the previous trial (temporal context). The analysis consisted of two steps. First, we performed PCA on the OFC population activity and used the first 12

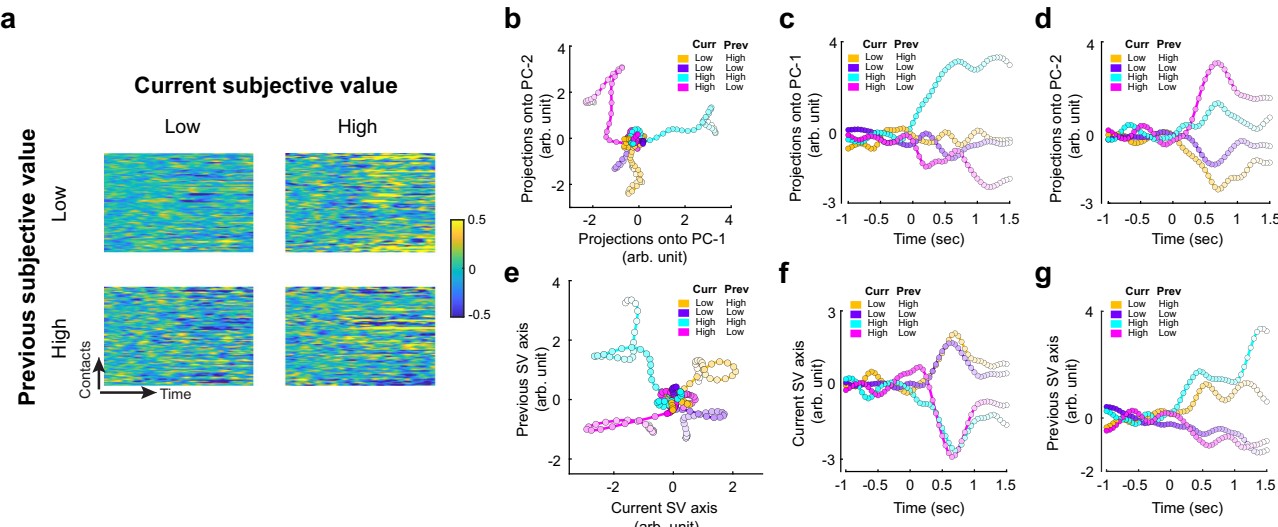

**Fig. 9 | State-space analysis reveals OFC activity trajectories in a low-dimensional value-context space. a** OFC activity sorted by condition. We sorted all trials from all contacts into four different conditions according to the magnitude of the current subjective value (high or low, median split for each subject separately) and the previous subjective value (high or low, median spilt for each subject separately). Each graph summarizes the average OFC activity timeseries of 166 electrode contacts. Each row represents the timeseries−from 1 s before stimulus onset to 1.5 s after stimulus onset−of a single OFC contact. **b–d** Principal component analysis (PCA) of OFC population activity reveals distinct, value- and context-specific activity trajectories. **b** Activity trajectories are plotted and color-coded with respect to the four conditions in the space of the first and second principal components. Color codes for the four conditions: [current subjective value, previous

subjective value] = [high, low] (magenta), [high, high] (cyan), [low, low] (purple), [low, high] (yellow). The horizontal and vertical axes represent the projections of OFC population activity onto the first principal component (PC-1) and the second principal component (PC-2) respectively. **c** PC-1 captures the subjective value from the previous trial. Here, we plot the activity trajectories on the PC-1 axis as a function of time. **d** PC-2 captures the subjective value of the current trial. Here we plot the activity trajectories on the PC-2 axis as a function of time. **e–g** Regression subspace analysis. **e** Activity trajectories are plotted in the space constructed of the current SV axis (horizontal axis) and the previous subjective-value axis (vertical axis). **f** Activity trajectories on the current SV axis plotted as a function of time. **g** Activity trajectories on the previous subjective-value axis plotted as a function of time. Source data are provided as a Source Data file.

PCs (accounting for 78.59% of the neural variance across electrode contacts) to construct a denoising matrix ($D$) used to obtain the orthogonal axes described next. Second, we projected the population activity onto two orthogonal axes−the axis of the current subjective value (horizontal axis) and the previous subjective value (vertical axis) −that were defined based on the regression coefficients of subjective value in GLM-1 (see *Regression subspace analysis* in *Methods* for details). The trajectories from the four different conditions again started at the same origin and quickly established their unique paths after stimulus onset (Fig. 9e) in a manner similar to the one observed in the first two components of the raw PCA. As expected then, the current subjective value regression axis could easily distinguish the current subjective value (Fig. 9f), separating high current subjective value (cyan and magenta) from the low current subjective value (purple and yellow). The emergence of this separation appears to begin within 500 ms after stimulus onset. The temporal context (previous subjective value) axis could easily distinguish the previous subjective value (Fig. 9g), separating the high (yellow and cyan) from the low (purple and magenta) previous subjective values. Finally, we found that these patterns remained when we varied the number of PCs for $D$ (Supplementary Fig. 20) and when no data smoothing was applied prior to the PCA (Supplementary Fig. 21).

In summary, these two analyses reveal the dynamics of OFC activity in representing the subjective value and temporal context and highlight the fact that context is a major source of organized patterns of activity in the OFC. At least while subjects are performing valuation tasks, value and context seem to be the major determinants of activity patterns in this region.

## Discussion

In this study, we used sEEG in human epilepsy patients to investigate the electrophysiological representation of subjective value and context, in humans−fundamental building blocks in the theory of

decision-making widely studied in non-human animals. Our data show that, as observed previously in animals, human subjective value signals show strong evidence of temporal context dependency. Indeed, our PCA results suggest that context is an even more significant determinant of activity pattern than simple value. Previous work has indicated that the human OFC represents the subjective value of rewards under immediate consideration: Gamma (30–80 Hz) and high-gamma activity (80–150 Hz) have both been shown to positively correlate with the subjective value of an offered reward[59]. We found that these signals were negatively influenced by the magnitude of rewards offered on previous trials, a form of temporal context dependency that has been observed behaviorally in humans and physiologically only in non-human primates. To our knowledge, no prior study has shown evidence of context-dependency in human electrophysiological signals encoding subjective value.

An important−and often less explored in previous studies−dimension of our data analysis was an analysis of signals at the single contact/electrode level. Many previous human intracranial studies have been forced to average across electrode contacts in order to report findings about a region of interest. The high quality of both our initial signals and the nature of our analytic pipeline, allowed us to both examine single-contact level data and region of interest averages. The single-contact level data allowed us to examine the local spatial distribution of both subjective value and context signals, independently, at the level of cortical subareas. Our data revealed that signals from many individual contacts in the lateral OFC encode a subjective value signal that is only weakly influenced by temporal context. In contrast, subjective value signals influenced by context were more common in the central and medial subregions of the OFC. In a similar vein, we found that the hippocampus and insula carried subjective value signals strongly influenced by temporal context at the single contact level.

Our contact-by-contact data also indicates that even in areas known to carry robust subjective value signals, only about 30% of the

recording sites carry those signals (in a statistically significant way). The observed patchy distribution of subjective value signals in the human agrees well with parallel work in the monkey conducted with much finer electrodes gathering signals at the single neuron level.

There is now widespread agreement that subjective value representations are broadly distributed in the mammalian brain. A series of influential meta-analysis studies examining fMRI data from human subjects have focused attention on two key areas: the ventral striatum and the ventromedial prefrontal cortex. Activity in these two areas is now widely associated with decision-making and subjective value and measurements of these areas are now often used as direct tools for assessing subjective value in humans. Interestingly, however, the focus on these two human brain areas stands in contrast to extensive electrophysiological work on value representations in non-human primates and to a lesser extent in rodents. Multiple electrophysiological studies in animals have focused interest on the OFC as critical source of subjective value signals in these species. This apparent mismatch between human and monkey data has been of some significance, and has raised the question of whether this mismatch reflects a technological difference between fMRI and electrophysiology or a species difference between humans and other mammals.

Recently, a small number of intracranial electrophysiological studies have begun to address this dichotomy by using electrophysiological tools to examine subjective value representations in humans. Studies on value-based decision-making in humans have now been conducted using electrocorticography (ECoG)[58] and sEEG in human epileptic patients[59,70,71]. Since the high-frequency components (the gamma and high-gamma bands) of the sEEG and ECoG signals are believed to correlate tightly with the single unit activity[61,72–77], it should be possible to use these tools to search for the human analog of monkey OFC value signals. Lopez-Persem and colleagues[59] for example, used a region-of-interest approach with sEEG recordings and showed unambiguous subjective value signals that were the human homolog of monkey value signals in that same region.

Our observations extend this earlier work from the level of a region of interest to the level of a single recording contact. This adds to a growing literature using sEEG or ECoG in humans that finds subjective-value representations in the medial (area 14) and central OFC (areas 11 and 13) in single-unit responses[71], in gamma and high-gamma activity[58,59], and in single-unit activity in the amygdala[70]. It should, however, be noted that the naming of the subdivisions of OFC are not consistent across studies. Here we have adopted the monkey-based convention used by Padoa-Schioppa and Cai[78] and Wallis[79] in describing the medial (area 14), central (areas 11 and 13), and lateral (area 47/12) OFC.

Our current understanding of the subjective valuation of rewards in humans is based on three different methods for eliciting subjective value—choice tasks, the BDM auction task where subjects indicate willingness-to-pay, and liking-rating tasks. At the behavioral level, earlier work highlighted inconsistencies between BDM task and choice task[80–83]. More recently, however, Lopez-Persem and colleagues[84] did find that different elicitations of subjective value (choice, rating, and effort tasks) tend to produce consistent results. However, at the neural level, there has been little discussion to date on how these different elicitation methods might differentially impact the subjective-value signals measured physiologically.

While we employed the incentive compatible BDM value elicitation method to assess a subject's subjective valuation for food items[70], most other studies have employed either the liking-ratings tasks under similar conditions[59,71] or asked the subjects to choose between different monetary lotteries[58]. We chose the incentive compatible willingness-to-pay approach presented here for two reasons. First, unlike the liking-rating approach, with the BDM it is in the subjects' best interest to provide their true valuation of the rewards and some research indicates that this yields more accurate estimates of

subjective value at the behavioral level[63]. Second, compared with choice tasks that offer two or more options, each trial presents only one object for evaluation, simplifying the interpretation of neural data to the representation of a single object. Despite these advantages of the BDM approach, however, it must be acknowledged that the cognitive and motivational processes associated with each method are not identical. How might these different elicitation methods impact subjective valuation? It will be important for future investigations to enrich our understanding of the neural representations for subjective value by characterizing the similarities and differences between these different methods as they impact subjective value signals in the brain.

Numerous studies in non-human primates have made it clear that the subjective value signals in the OFC are strongly influenced by context. Padoa-Schioppa[28], for example, showed that the firing rates of single OFC neurons in monkeys are strongly influenced by the temporal context. These non-human primate signals appear to be affected by the range of subjective value experienced in the recent past[15,28,85,86]. It remains an open question as to whether the human OFC activity exhibits this same property. Another important and open question is how these relative-value signals in OFC contribute to choice behavior. While studies had begun to show the effects of range on choice behavior[31], the links between OFC relative-value signals and range-affected choice behavior remain unclear.

Our data show that recently experienced rewards influence the activity of human subjective value neurons in these same areas. We found that the reward delivered on the preceding trial effectively down-adapted the signal observed from the human OFC. This is a finding compatible with the standard models of this process, both range adaptation models and divisive normalization models, which are aimed at describing the influence of recent temporal context on OFC firing rates. We stress that the fact that our linear regression analysis suggests a subtractive relationship between current and previous reward should not be interpreted as specifically supporting a subtractive relationship in the neuronal data. Linear regression is constrained to always represent divisive relationships as subtractive. Were the true relationship divisive as in the divisive normalization models, it would be expected to appear subtractive upon linear regression as used here. For this reason, we must be silent about the true form of the representation.

Although we found context-setting signals—the representations of previous subjective value—in the OFC and other brain regions, there are several alternative interpretations of the previous-value representations worth discussing. First, it is possible that the previous-value signals simply reflect the autocorrelation in the neural signals. To address this issue, we performed additional analyses and found that in general, signal autocorrelation did not significantly change the previous-value findings (Fig. 3f, g; Supplementary Figs. 11, 12). Second, it is possible that the BDM bidding task induces a perceived cost at trial $t$ and is then considered in trial $t+1$. We think this might be less likely to occur as we did not realize the auction after each trial and because the actual paid cost in a BDM auction is determined randomly only at realization at the end of the experiment. The subjects were told that, at the end of the experiment, only one trial would be chosen at random and realized. Such design is an attempt to minimize the impact of previous decisions and is adopted by many decision-making experiments and neuroeconomic experiments[87]. Indeed, when reward feedback is given after each trial, previous decision outcomes are more likely to incur a mental cost that would make people to want to break even (e.g., house money effect[88]). People are also more likely to maximize expected value when reward feedback is given after each trial[89]. Finally, we wish to acknowledge that context dependency here can also be described as "adaptation" or "adaptive coding". We regard context dependency as the super category, and spatial and temporal adaptation as subcategories. We elected to refer to the effect we saw as context dependency in this more general sense. In the mid-1980s,

Ohzawa et al. [26,90] used the formulation 'adaptation' in their classic studies of V1 neurons where neuronal activity adapts to the mean contrast level of the visual stimulus. This phenomenon was termed contrast gain control. Because the spatial and temporal scales are the key factors for determining the mean contrast level, adaptation naturally is affected by both the spatial and temporal context of the environment[91]. Following that usage of adaptation, many other labs who went on to describe center-surround organizations as relying on a similar mechanism introduced the notion that the spatial context terms could in parallel be called spatial adaptation. Indeed, in our work and others context dependency is generally referred to as being made up of spatial and temporal adaptation[27,28,31].

Over the last decade there has been increasing interest in aggregating information from large populations of neurons in macaques and rodents into high dimensional datasets that can then be analyzed at the population level[92–94]. Of particular interest has been the extraction of the temporal trajectories—in low-dimensional space—neuronal activity takes in these high dimensional data-spaces in valuation and decision-making tasks. In this report we extended those approaches to the study of human sEEG signals. Our results both validate and extend this earlier work in non-human primates. Our unbiased PCA revealed that during our task the population begins each trial at a common starting point and then evolves toward a representation whose primary properties are a representation of reward context and current offer, with the suggestion of context being an even larger signal than value. Our regression subspace dimensionality reduction analysis[68] further confirmed and extended this finding, revealing that OFC population dynamics formed distinct trajectories according to the subjective value and context-setting signals. These trajectories serve as another point of contact with monkey data, reinforcing the similarities of these two species in the OFC.

Previous monkey studies have also provided some sense of how subjective value and context signals are distributed in the monkey brain. While subjective value signals are observed robustly in areas like the monkey OFC, it is important to note that not all OFC neurons show these signals. Estimates from Rich and Wallis[61] made in the monkey suggest that only about 30% of channels in the OFC carry a subjective value signal. They found that after stimulus onset, the peak percentage of OFC channels whose high-gamma activity represented expected reward size was about 20%. After a reward was delivered, the peak percentage was about 40% in representing the type of reward the animals received. Interestingly, we observed a similar result: in high-gamma activity, about 30% of our OFC electrode contacts showed evidence of either the subjective value signals, or the context setting signals (subjective value of the previous trial), or both. This seems to show excellent agreement between humans and monkeys.

Our examination of context dependency, however, does seem to suggest a difference between humans and monkeys. We observed that in some areas, like the central and medial OFC, individual contacts reflected either subjective value signals or context setting signals. Very few contacts represented both the subjective value and context setting signals. This is an observation that has not been widely reported in the monkey. While it will be important to confirm these findings, this does raise the possibility that human context setting signals may be distinctive in some way. A related point is the intermixing of positive and negative encoding of subjective value. Our single-contact analysis showed that, for both the subjective value and temporal context, OFC contacts showed either positive or negative correlation with these variables. This is in part consistent with monkey electrophysiology studies showing intermixed encoding of subjective value in the OFC—some neurons positively correlated with subjective value, while others showed negative correlation[6]. As these positive and negative encoding neurons are similar in number, multivariate decoding analyses that take these neurons or populations as data should be able to readout these intermixed subjective-value signals. Indeed, human fMRI studies

using multivoxel decoding analysis found subjective value signals in the central OFC[48] and context-dependent responses to reward outcomes in the ventromedial prefrontal cortex[30,56]. Our single-contact results further suggested the possibility of decoding subjective value and context-setting signals in human fMRI with multivoxel decoding approach.

Results from non-human primates also may shed light on how broad-band sEEG signals might be expected to behave. For example, it has been suggested that low-frequency activity in the alpha band may be involved in modulating inputs from task-relevant and task-irrelevant brain regions[95]. In studies of reward representations, An et al. [96] found that in non-human primates, reward expectation increased single-unit firing rates in the primary motor cortex but decreased alpha (8–14 Hz) oscillatory power. Given that single-unit firing rates often positively correlate with high-gamma power[76], this finding suggests that the high-frequency power (e.g., high-gamma activity) in M1 may positively correlate with reward expectation, while low-frequency power, such as alpha power, negatively correlates with reward expectation. This is similar to what we found where the encoding directions for subjective value in the low-frequency activity were reversals of those in the high-frequency activity.

One potential explanation for the encoding directions of subjective value in the alpha band is the inhibition of the previous subjective value. Increase in alpha oscillations have been shown to reflect inhibitory activity in circuits associated with attention, perception, and working memory[97–104]. Hence, it is possible that the positive correlation of alpha activity with the previous subjective value reflects the inhibition of information about past subjective value when the subjects evaluate a snack food item in the current trial. Similarly, the negative correlation of alpha power with the current subjective value may reflect increased attention to the food item in the current trial. Our findings also suggest the involvement of alpha oscillations in modulating visual attention during value-based decision-making[105]. Previous fMRI studies found that value signals in the vmPFC are modulated by visual attention[106]. An open question, therefore, is to investigate whether and how alpha oscillations in the OFC, and other value-related regions, change in a free-choice paradigm where different options are simultaneously presented and the subjects are free to look at these options before making a decision.

Our results were consistent with the view that there is a close relationship between high-frequency oscillations (30–150 Hz) in LFP and the fMRI BOLD signals. Our results showed the involvement of gamma (30–80 Hz) and high-gamma (80–150 Hz) activity in the representation of subjective value. This frequency range (30–150 Hz) coincides with previous observations that "LFPs were often dominated by stimulus-induced and usually stimulus-locked fast oscillations in the range of 30–150 Hz"[107]. Given that BOLD signals have been found to be better described by the LFP than by multi-unit activity, it is possible that the gamma and high-gamma findings here would be observed in BOLD signals in the human OFC had there been the same spatial resolution or no signal loss due to the susceptibility artifacts in BOLD signals in this brain region. In brain regions associated with subjective value representation that do not suffer BOLD signal loss, we would expect the gamma and high-gamma activity there to represent subjective value. Indeed, we found that many other brain regions also represented the subjective value in high-gamma activity. Among them, the insula and hippocampus stood out because evidence for the past and present subjective-value representations were found at the group level (averaged across electrode contacts) and at the level of individual contacts in those regions.

One limitation of our task design is motor-related confounds. Although we aimed to minimize motor confounds by temporally separating the evaluation and the response stages, the subjects could in principle have prepared their motor responses during the evaluation stage given that the response matrix was the same throughout the

experiment. Future investigations could address this issue by not specifying the motor-response mapping until the response stage. For example, one can design a visual analog scale for bid value and could randomly, from trial to trial, vary the presentation of bid value (left-to-right indicates low-to-high value in half of the trials and high-to-low value in the other half).

Another important limitation for any sEEG study is the sparse and heterogeneous coverage of the electrodes. The decision about where to implant electrodes is, rightfully, an entirely clinical decision, but as a result the spatial coverage we can achieve is sparse and heterogeneous, both within and across the subjects. This limitation poses challenges to both within- and between-subject (group-level) statistical inference and the interpretation of null results. On the one hand, null results can signal that a region is not involved in certain tasks or computations. On the other hand, the null results could be driven by sparse or inefficient coverage. In the context of our study, it is insufficient to conclude, for example, that OFC does not represent subjective value on a particular subject based on the null results from his or her OFC contacts. It is possible that his or her OFC contacts are not in the right spot—regions in the OFC that represent subjective value. One possible way to address this issue is the development of distributed, anatomically realistic source modeling of LFP data[108]. Future studies need to explore this direction and to examine its feasibility and value in contributing to the interpretations of LFP signals in human sEEG experiments.

Sparse and heterogeneous coverages also raise two important questions. First, is it possible that the three regions with the strongest value signals (OFC, hippocampus, and insula) were also the regions with the highest number of contacts? We think that while this is possible, we also saw evidence against this conjecture. For example, the IPS had 62 contacts, and yet it did not show significant correlation with either the current or previous subjective value at the group level. By contrast, the PCC and amygdala had 31 and 30 contacts respectively, and in these two brain regions we found significant results on either the current subjective value (amygdala) or the previous subjective value (PCC). Therefore, it is not entirely the case that brain regions with more contacts had the strongest value signals. Second, is it possible that subjects with more contacts in a given region contribute more heavily to aggregate findings for that region? We think this is less likely to be the case. For OFC, no single subject contributed to more than 7.23% of the contacts. The median fraction of contacts contributed by a single subject was 5.12% (range: from 0.6% to 7.23%). To illustrate, in Supplementary Fig. 22, we plot the subject-by-subject results in the OFC and present them in an order according to the number of electrode contacts from each subject. In summary, we wish to point out that these remain important questions and need to be more systematically addressed as sEEG studies mature as a field.

Many of the behaviors in humans and animals are affected by the context of our recent experience. Characterizing the representations of context at the computational, algorithmic and neural implementation levels, therefore, is essential to understanding a wide array of cognitive functions. Using human intracranial electrophysiology, we found several distinct features of context-dependent representations for the subjective valuation of rewards. At the computational and algorithmic levels, temporal context—recent history of rewards—was represented in a manner predicted by existing models like divisive normalization and range adaptation. At the neural implementation level, we found that the current reward value and the context were represented by distinct electrode contacts in the OFC, insula, and hippocampus. These findings suggest that contextual adaptation is implemented through distinct, large-scale neuronal populations that separately represent current and past information about reward value.

## Methods

### Participants

Twenty patients with drug-resistant epilepsy participated in this study (9 males; aged 16–51 years; average: 29.2 years). Patients had been implanted with multi-contact depth electrodes and were undergoing intracranial monitoring in order to identify seizure onset regions. Each patient was implanted with 7–14 electrodes. The decision to implant the electrodes and their location was driven solely by medical considerations. The study was approved by Taipei Veterans General Hospital Institutional Review Board. To learn more about the patients' decision-making behavior in relation to the normal population, we collected behavioral data from 35 healthy subjects in the normal population not treated for epilepsy (18 males, aged 20–27 years; average: 22 years) performing the same task. This part of the study was approved by National Yang Ming Chiao Tung University Institutional Review Board. Informed consent was obtained from each participant before participation.

### Behavioral task

The subjects performed a version of the BDM auction task during sEEG recording. The task was programmed in MATLAB (The MathWorks Inc.) using the Psychophysics Toolbox Version 3[109,110]. They were asked to refrain from eating for at least two hours before the start of the experiment. Prior to the BDM task, the subjects received 200 New Taiwan Dollar as an endowment to purchase food items.

The BDM task consisted of 8 blocks of 25 trials each. One hundred snack food items were introduced and each food item was presented twice in the experiment. In each trial, a snack food item was presented and the subjects were instructed to bid—his or her maximum willingness-to-pay—for the snack food item. A trial started with a fixation cross presented in the center of the screen for 1 s. Following the fixation, a snack food item was presented on the screen until subjects clicked on the mouse button to signal that she or he was ready to indicate his or her willingness-to-pay. The subjects could take as long as they wanted to indicate their readiness. Trials where the RT was three standard deviation away from the mean RT of the subject were excluded from further analysis (usually less than 1% of trials), as they could very well indicate disruptions of the experiment outside of the experimenters' control (e.g., visits from clinicians, nurses, and/or staffs). After making the mouse click, a 1 s fixation period followed. After the fixation period, the subjects would see a price matrix from 0 to 200 in steps of 10 on the computer screen (Fig. 1a). The subjects' task was to move the cursor to the price box that reflected the most she or he was willing to pay for that food item. Subjects could take as much time as they desired to select one of these boxes with a mouse click but were encouraged to respond within 2 s. To give the subjects an idea of time, the box where the cursor was at would turn blue within 2 s after the price matrix box appeared. After 2 s, the box where the cursor pointed at would turn red. After clicking on the desired price box, a brief visual feedback on the selected price (willingness-to-pay) was shown (0.5 s), which was then followed by a variable inter-trial interval (1, 1.5, or 2 s).

In this task design, a single trial therefore consisted of two stages: an evaluation stage followed by the response stage. During the evaluation stage (when the food item was presented) the subjects were instructed to take time and think over how much money they were willing to pay for the food. By contrast, during the response stage (when the price matrix was shown) the subjects were instructed to indicate willingness-to-pay as quickly as possible. The reason for implementing the two-stage design was to temporally localize the valuation signals we hoped to observe. Since valuation of the food item and the motor response to indicate willingness-to-pay were temporally separated by our task, we hoped that the impact of motor-related confounds introduced when subjects indicated willingness-to-pay could effectively be minimized.

After all trials of the BDM task were complete, one trial was randomly selected and realized according to the rules of the BDM auction. The rules are as follows. Let $x$ be the bid made by a subject for a food item. A random integer $y$ is drawn from a discrete uniform distribution ranging from 10 to 200 with interval of 10. If $x \geq y$, the subject would buy the food item at a price equal to $y$. If $x < y$, the subject would not get the food and would keep the endowment. These widely used rules establish a situation whereby the optimal strategy for the subjects is to bid exactly the maximum amount that they are willing to pay for the item. If they underbid for an item, the subject may lose the opportunity to purchase the item later at a still highly desirable price. If they overbid, they risk being forced to purchase the item at an undesirable price. Only by stating the exact maximum price at which they would purchase the item can they achieve the optimal result. The BDM rules and the consequences were informed to the subjects before the BDM task so that they knew that the best strategy was to bid exactly what they are willing to pay for the item.

### Electrophysiological recordings
Patients were implanted with 0.86 mm diameter depth electrodes (Ad-Tech, Racine, WI, USA) that were arranged into strips with 6, 8, or 10 contacts (2.29 mm in contact length) and 4–8 mm (most strips: 5 mm) separation. One of the electrode strips was 1.12 mm in diameter with six contacts (2.41 mm in contact length) and 5 mm separation in between neighboring contacts. Recordings were obtained simultaneously from the scalp and depth electrodes while the patients performed the task. Data was collected using the Natus Quantum system (Natus Medical Incorporated). Sampling rates were 2048 Hz with an 878 Hz low-pass filter. During recording, all the electrodes were referenced to the scalp PFz electrode or an intracranial contact located in the white matter. Details on the recording sites—MNI coordinates of the electrode contacts—presented in the current study can be found in the Supplementary Tables 9–16.

### MRI acquisitions
For each patient, T1-weighed structural MRI images were collected on a 1.5 T Signa HDxt scanner (GE Healthcare, Milwaukee, WI, USA) before and after the surgery for electrode implantation. The MR images were taken along the axial plane using a fast spoiled gradient-recalled echo sequence (axial slice thickness = 1 mm; TR = 10.02 ms, TE = 4.28 ms, TI = 0 ms, flip angle = 15°, bandwidth = 31.2 kHz, matrix = 256 × 256, FOV = 256 × 256 mm).

### CT acquisitions
CT images were used in conjunction with T1-weighted MRI images for transforming the anatomical location of the electrode contacts onto the standard MNI space. CT images were acquired using Philips Brilliance 64 CT scanner with the following parameters: 64 slices, rotation duration of 1 s with coverage of 16 cm per rotation, 60-kW generator (512 × 512 matrix), 120 kV, 301 mAs, and axial slice thickness of 1 mm.

### sEEG data analysis
Below we describe the sEEG data analysis pipeline in detail. A summary of the sEEG data analysis pipeline can be found in the Supplementary Fig. 7.

### sEEG preprocessing
EEG data were preprocessed and analyzed by EEGLAB[111] (version 2022.1) and ERPLAB[112] (version 9.00) in MATLAB in the following steps. First, a digital band-pass filter from 0.5 Hz to 250 Hz and a 60 Hz notch filter were applied to the EEG data at the single contact level—including the scalp and the sEEG dataset. Second, the scalp EEG dataset were separated from the sEEG dataset. Third, the sEEG data for each electrode contact were re-referenced to the average of the two neighboring contacts[113]. The scalp EEG data were re-referenced to the left

and right mastoid. Fourth, in order to remove eye-movement-related activity from the sEEG data, we proceeded in two steps, which were separately applied to both the scalp EEG and sEEG data. First, we performed PCA on the data and, by keeping the PCs that explained 95% of data variance, we effectively reduced the dimensions of the data. Second, we performed ICA using data based on the PCs that explained 95% of data variance. Eye-related activities were first identified by inspecting the independent components (ICs) of the scalp EEG data. Once an IC with ocular artifacts was identified, we checked whether there was a corresponding IC in the sEEG data. We note that because of the first step, the number of ICs identified in the second step would be greatly reduced compared with directly performing ICA on the raw sEEG data. And since eye-movement artifacts are mostly likely very strong artifacts, this approach in principle would not compromise the ability to identify these ocular artifacts. Rather, it saved us significant computation time.

The epoch for each trial (trial epoch) started 1.5 s before the onset of the food stimulus and ended 2 s after stimulus onset, with a pre-stimulus baseline correction where the average activity of the 1.5-s time window before the onset of the food stimulus was treated as the baseline. Trial epochs with interictal spikes were identified through visual inspection and were excluded from further data analysis. We referred to the trials with no interictal spikes as the valid trials.

### Time-frequency analysis
After preprocessing, a time frequency analysis was performed using a wavelet transform, estimating spectral power from 4 to 200 Hz for each epoch with full-epoch length single-trial baseline correction[114]. After time-frequency analysis, the epoch of the timeseries data started at 1 s before stimulus presentation and ended at 1.5 s after stimulus presentation with a 10 ms resolution. We note that in our data preprocessing, the EEG epochs were slightly longer (1.5 s before to 2 s after onset) so that we can apply time-frequency analysis upon them without zero-padding. For each frequency bin separately, we computed the mean power (averaged across time within the 1-s time window before the onset of the food stimulus) and used it as the baseline. We then subtracted baseline from the timeseries data of spectral power. The timeseries of the power data from the high-gamma (80–150 Hz), gamma (30–80 Hz), beta (13–30 Hz), alpha (8–12 Hz), theta (4–7 Hz) bands for the epoch were further extracted for the GLM analysis described below (see *General linear modeling of brain activity* below). At each frequency band, the corresponding timeseries data had 251 time points (a 2.5-s time window with a 10-ms resolution).

### Identifying the anatomical locations of electrode contacts
To identify the anatomical location of electrode contacts across different subjects, we transformed the electrode contact location from the subject's native space onto standard Montreal Neurological Institute (MNI) space. To do that, we used three sets of brain images collected from each subject: the T1-weighted image prior to electrode implementation (pre-T1), the T1-weighted image after electrode implementation (post-T1), and the CT image after electrode implantation (post-CT). Our goal was to transform the CT image to MNI space. The reason we used the CT image to identify the electrode contact coordinates in the standard space was because the CT image, compared with T1-weighted image, suffers less distortion and therefore allows for more accurate mapping of the contact location. The transformation was performed using SPM12 (Wellcome Trust Center for Neuroimaging, London, UK; https://www.fil.ion.ucl.ac.uk/spm/) and proceeded in the following three steps. First, the post-CT image was aligned to the post-T1 image with 4th degree B-Spline interpolation. Second, both the post-T1 image and the realigned post-CT image were aligned with the pre-T1 image, also with 4th degree B-Spline interpolation. Finally, the pre-T1, the realigned post-T1 and post-CT images were transformed to the standard MNI space (1 mm isotropic voxel

size). The location of each contact in MNI space was obtained through the post-CT images in MNI space.

## Identifying the OFC electrode contacts

We used the Harvard-Oxford probabilistic atlas available in FSL[115] (version 6, https://fsl.fmrib.ox.ac.uk/fsl) to identify the electrode contacts in the OFC. An electrode contact was identified as an OFC contact when the probability of its MNI coordinates being in the OFC was larger than 1%. In addition, because some contacts were situated at the borders between the posterior section of the OFC and the anterior insula, we decided to exclude the contacts that had a higher probability of being in the insula than being in the OFC.

## Visualizing anatomical locations of electrode contacts

To visualize the anatomical location of electrode contacts across different subjects, we used the MNI coordinates of the contacts and plotted them in the standard MNI brain template. To show the electrode contact location, we used Fa-Hsuan Lin's toolbox (https://github.com/fahsuanlin/fhlin_toolbox)[108] to generate the brain images shown in Fig. 2a, Fig. 4a, Fig. 5a; 7a–c; Fig. 8a–d by superimposing the electrode contact location onto the surface-based MNI template that depicts the gray/white matter boundary. The MNI template used by the toolbox came from FreeSurfer[116]. As a result, some contacts can appear to be outside of the brain where in fact they are not. We used the Harvard-Oxford probabilistic atlas to verify that these contacts are in fact inside the brain. Information about the MNI coordinates of all the electrode contacts mentioned in the paper are in the Supplement (Supplementary Tables 9–16). To show the contact location on the example subject (Fig. 2d), we used FSLeyes in FSL[115] (version 6) to generate the brain images in Fig. 2d.

## General linear modeling of brain activity

For each contact, after time-frequency analysis, we obtained for each trial a time-series data of spectral power at a particular frequency band. Here we use high-gamma band (80–150 Hz) as an example and we refer to the power of high-gamma as high-gamma activity, as in ref. 61. To examine subjective-value representations, we performed the following GLM analysis. First, we set up a GLM for each time point within the epoch separately. Here the data—a vector of length $N_{trials}$ where $N_{trials}$ is the number of valid trials a subject performed in the BDM task—are the frequency-specific power obtained from time-frequency analysis (see sEEG preprocessing for descriptions on valid trials). We implemented five different but similar GLMs. The first GLM (GLM-1) was the main GLM, and the rest were slightly different versions of GLM-1 in order to test the robustness of GLM-1's results on subjective-value representations (Fig. 3). In GLM-1, we implemented a constant term, a regressor for the subjective value of the current trial, and a regressor for the subjective value of the previous trial. In GLM-2, we performed the analysis in two steps. First, we implemented a model with the constant term and the current subjective value regressor. Second, we used the residuals from the first step as data and implemented a model with the constant term and the previous subjective value. In GLM-3, we implemented the same model as GLM-1 except that we only considered trials where the subject's willingness-to-pay was greater than zero dollars (excluding zero-bid trials). In GLM-4, the subject's RT was added as a regressor, along with the constant term, the current subjective value, and the previous subjective value. In GLM-5, we added the subjective value of the food option encountered two trials back as a regressor, along with the constant term, the current subjective value, and the previous subjective value. In GLM-6, in addition to the current and previous subjective value regressors, we added the spectral power of the previous trial as a regressor. That is, for each time point separately, the spectral power of the same time point from the previous trial was used as the regressor. Finally, to examine whether the length of the inter-trial interval (ITI; 3 possible ITIs at 1 s, 1.5 s, and 2 s) affected

the current and previous subjective value results, we estimated GLM-1 separately for each ITI (1 s, 1.5 s, 2 s). We used the *fitlm* function in MATLAB to perform the GLM analysis.

## Group-level permutation test (across electrode contacts)

This analysis was used for Figs. 2b, 3, 4b, 5b, 6b, 7 (middle graphs), and 8 (middle graphs). The data is an $N_{contacts} \times N_{time}$ matrix of estimated regression coefficients. $N_{contacts}$ denotes the number of electrode contacts in a region of interest (e.g., OFC), and $N_{time}$ denotes the number of time points within the trial epoch. Using this data, we computed the timeseries of the $t$ statistic (across electrode contacts). Hence, the $t$-statistic timeseries is a $1 \times N_{time}$ matrix. To compute the $t$ statistic, at each time point we computed the mean regression coefficient (across contacts) and divided it by its standard error. We then transformed the $t$ statistic to the $z$ statistic. To correct for multiple comparisons across time points, we performed a permutation test with TFCE as the test statistic[64,65]. In each permutation, we randomly assigned a label of 1 to half of the contacts and −1 to the other half. Note that for each permutation, this procedure—assigning one to half of the contacts and −1 to the other half—was applied to all the time points of the trial epoch. For each time point separately, we then performed a linear regression analysis where data is the actual regression coefficients, and the regressor was the randomly permuted label. This gives us a $t$ statistic for the regressor at each time point. The $t$ statistic was transformed to the $z$ statistic, and using the $z$ statistic we computed the TFCE statistic ($E = 2$, $H = 2$) for each time point. The TFCE statistic summarizes the strength of spatially or temporally extended signals, i.e., spatial or temporal clusters. A major advantage of using the TFCE statistic is that we do not need to specify a cluster-forming threshold in order to identify spatial or temporal clusters of activation. The TFCE statistic is defined as the sum of the scores of all supporting sections underneath the $z$ statistic of a particular time point (temporal cluster) or spatial location (spatial cluster). The score of each supporting section is its height (raised to some power $H$) multiplied by its extent (raised to some power $E$). The $E$ and $H$ parameters therefore control the impact of cluster extent and height respectively. The TFCE statistic therefore is a weighted sum of the entire local clustered signal[64]. As a result, after each permutation, we obtained a timeseries of the TFCE statistic, which we then used to identify the maximum (or minimum) TFCE statistic. Since the TFCE was computed based on the $z$ statistic, for positive effects (positive $z$ statistic) we looked for the maximum TFCE and for negative effects (negative $z$ statistic) we looked for the minimum TFCE. After 10,000 permutations, we obtained the null distribution of the maximum (or minimum) TFCE and used it to determine the critical region ($p < 0.05$, familywise error corrected). The TFCE statistic at each time point was then evaluated with respect to the critical region: if the TFCE statistic fell within the critical region, we would conclude that it was statistically significant. Otherwise, it was assessed as not significant.

## Test of symmetry in the data distribution

An important assumption for the permutation test is the symmetry of the data distribution[65]. For the group-level permutation tests described above, we had two datasets, one containing the regression coefficients of the current subjective value, and the other for the previous subjective value. Taking the current subjective value as an example, the data is an $N_{contacts} \times N_{time}$ matrix where each element is the regression coefficient of the current subjective value of a particular electrode contact at a particular point in time in the trial epoch. Each time point consists of a distribution of the regression coefficients across electrode contacts. Our goal was to examine whether this distribution ($N_{contacts} \times 1$ matrix of regression coefficient) was symmetrical. Therefore, for each time point separately, we computed its corresponding sample skewness (Pearson median skewness). To test symmetry, we used the bootstrap method (resampling with

replacement) so as to obtain the distribution of the sample skewness and use it to construct the 95% confidence interval of sample skewness. If the 95% confidence interval did not include 0, we would conclude that the data distribution was symmetrical.

Taking the OFC high-gamma activity as an example: In order to obtain the distribution of the sample skewness, first, we resampled with replacement the OFC contacts (166 in total) 10,000 times. This gave us, for each resampled dataset, a time series of regression coefficients, separately for the current subjective value and the previous subjective value, from the 166 resampled contacts. Second, for each time point separately within the time series, we computed the sample skewness (Pearson median skewness) of the regression coefficients across the resampled OFC contacts. Third, with 10,000 resampled datasets, we obtained, for each time point separately, a distribution of sample skewness that we used to construct the 95% confidence interval. Finally, using the 95% confidence interval, we were able to test whether the data distribution was symmetric at each time point separately. We found that the sample skewness did not differ significantly from 0 in the majority of time points, for each brain region, each frequency band, and for each regressor of interest (current subjective value and previous subjective value). The results are shown in the supplement (Supplementary Figs. 23–25). We observed a significant positive skewness for the current subjective value in OFC high-gamma activity approximately at 600–700 ms after stimulus onset (left graph, Supplementary Fig. 23a). On the one hand, this does raise concern for the permutation test regarding the current subjective value at this particular 100-ms time window. On the other hand, we also observed that the time window of activity that significantly correlated with the current subjective value (from ~400 ms to 1500 ms after stimulus onset; Fig. 2b) far extended this 100-msec time window. In other words, the significant current subjective-value representations included many time points where the skewness was not significantly different from 0. On this ground, we concluded that the violation of the symmetry assumption observed here should not change the overall conclusion that OFC high-gamma activity represented the current subjective value.

**Group-level permutation test (across electrode contacts) in the time-frequency space**

This analysis was used for Fig. 6a. The analysis logic is similar to that described in *Group-level permutation test (across electrode contacts)* described above. The main difference is the dimensionality of the dataset. Here the dataset included 166 OFC contacts across 20 subjects. In the GLM, we regressed the power against the current and the previous subjective value for each contact in each time-frequency point. As a result, for the current subjective value and previous subjective value separately, we obtained information about the regression coefficient (which we also referred to as the beta value) in a three-dimensional space (time, frequency, electrode contacts). Let $N_{freq}$ denote the number of frequency points (from 4 Hz to 200 Hz), $N_{time}$ denotes the number of time points within the trial epoch, and $N_{contacts}$ denotes the number of electrode OFC contacts. To correct for multiple testing across time points at the group level (across all contacts), we performed permutation test with threshold-free cluster enhancement (TFCE)[64,65] as the test statistic. In each permutation, we randomly assigned a label of 1 to half of the contacts and −1 to the other half. For each time-frequency point separately, we performed a linear regression analysis (see *General linear modeling of brain activity* above) where data was the actual beta values and the regressor was the randomly permuted label. This would give us the $t$ statistic of the regressor at each point in the two-dimensional time-frequency space. The $t$ statistic was then transformed to the $z$ statistic and based on the $z$ statistic, we computed the TFCE statistic ($E = 1$, $H = 2$) at each point in the time-frequency space. As a result, we obtained a two-dimensional map of TFCE and identified the maximum (or minimum) TFCE. Since

the TFCE was computed based on the $z$ statistic, for positive effects (positive $z$ statistic) we looked for the maximum TFCE and for negative effects (negative $z$ statistic) we looked for the minimum TFCE. After 1000 permutations, we obtained the null distribution of the maximum (or minimum) TFCE which we then used to determine whether each point in the time-frequency space was significant ($p < 0.05$, familywise error corrected). For other brain regions, 500 permutations were performed. The permutations were very computationally intensive and required much time. In general, we did not find the results to differ much between 500 and 1000 permutations. To save time we elected to use 500 permutations for the results shown in the Supplement (Supplementary Figs. 17, 18).

**Individual-level permutation test (for individual electrode contacts)**

This analysis was performed for each individual electrode contact separately and was used for Figs. 2c, d, 4c, 5c, 6c, 7 (right graphs), and 8 (right graphs). At each individual contact, the data is a $N_{trials} \times N_{time}$ matrix of brain activity where $N_{trials}$ denotes the number of valid trials and $N_{time}$ the number of time points within the trial epoch. Here, brain activity is referred to as the power of a particular frequency band (e.g., high gamma for 80–150 Hz) after time-frequency analysis. For each time point separately, we regressed brain activity ($N_{trials} \times 1$ matrix) against the current subjective value and the previous subjective value (GLM-1). This gave us a timeseries of regression coefficients for each regressor ($1 \times N_{time}$ matrix) and their corresponding $t$ statistics. To correct for multiple testing across time points, we performed a permutation test with TFCE[64,65]. At each permutation, we randomly permuted the trial label ($N_{trials} \times 1$ matrix) of the design matrix ($N_{trials} \times 2$ matrix where the current subjective value and previous subjective value were the two regressors). Note that for each permutation, the same permutation was applied to all the time points. We then regressed the data ($N_{trials} \times 1$ matrix), for each time point separately, against the permuted design matrix. This gave us a $t$ statistic of the regressor at each time point. The $t$ statistic was transformed to the $z$ statistic, and using the $z$ statistic we computed the TFCE statistic ($E = 2$, $H = 2$) for each time point. As a result, we obtained a time series of TFCE statistic and identified the the most extreme (maximum or minimum) TFCE across time. Since the TFCE was computed based on the $z$ statistic, for positive effects (positive $z$ statistic) we looked for the maximum TFCE and for negative effects (negative $z$ statistic) we looked for the minimum TFCE. After 10,000 permutations, we obtained separately the null distribution of the maximum and minimum TFCE that we used to determine the critical region ($p < 0.05$, familywise error corrected). If the TFCE statistic of a time point was inside the critical region, it would be labeled as statistically significant. Otherwise, it was labeled as not significant.

**State space analysis**

To study how the dynamics of high-gamma activity in the OFC as a whole represented subjective value, we performed two complementary state space analyses, one based on the PCA and the other based on a regression subspace analysis[68]. The data preparation for both analyses was identical and was performed in the following sequence. For each electrode, we first smoothed the high-gamma timeseries data for each trial (Gaussian time window = 400 ms). We also performed the same analysis with no smoothing applied, and with 100 ms, 200 ms, and 300 ms Gaussian time window (Supplementary Fig. 19). The timeseries data started from 1 s before the onset of the food stimulus and ended 1.5 s after stimulus onset. Second, we computed the average timeseries (across all trials) and subtracted it from the timeseries of each trial. Third, we sorted the trials into four conditions according to the magnitude of the subjective value of the current trial (high or low, median split) and the magnitude of the subjective value on the previous trial (high or low, median split). The

medians were obtained based on the corresponding subject's willingness-to-pay data. The four conditions therefore are [current subjective value, previous subjective value] = [high, low], [high, high], [low, low], [low, high]. Fourth, we computed the average timeseries of each condition. Fifth, for each condition, we organized the average timeseries data of all electrode contacts as a two-dimensional matrix of size $N_{contacts} \times N_{time}$ where $N_{contacts}$ is the number of electrode contacts in the OFC across all subjects ($N_{contacts} = 166$) and $N_{time}$ is the number of time points within the trial epoch ($N_{time} = 251$). We denote this condition-specific matrix $X_c$. Finally, we collapsed the four two-dimensional activity matrices (one from each condition) such that the final dataset for the subsequent analyses (PCA-based analysis and regression subspace analysis described below was a two-dimensional matrix of size $166 \times 1004$ which included all data aggregated together before the initial PCA was performed. We denote this data matrix $X$.

## PCA-based analysis

We performed PCA on the prepared dataset described above using the *pca* function in MATLAB. The feature dimensions were the electrode contacts, and the observations were the time points within the trial epoch. We then projected the activity matrix of each condition onto the first two PCs, resulting in four different trajectories (timeseries) in the PC space. We then plotted the trajectories and color coded them (Fig. 9b–d).

## Regression subspace analysis

We first performed PCA on the prepared dataset to denoise the data. In the main text, the prepared dataset was the smoothed high-gamma timeseries data (Gaussian time window of 400 ms). We also performed the analysis with no smoothing applied (Supplementary Fig. 21). The number of PCs ($N_{pc}$) selected to construct the $D$ was 12. We also performed the same analysis described below with number of PCs being 2 and 20 (Supplementary Fig. 20). The $D$ was constructed

$$D = \sum_{i=1}^{N_{pc}} PC_i PC_i^T \qquad (1)$$

where $PC_i$ is the $i$-th principal component and is a column vector of size $N_{contacts}$. The resulting $D$ is a $N_{contacts} \times N_{contacts}$ matrix. The denoised data $X_{pca}$ is obtained according to

$$X_{pca} = DX. \qquad (2)$$

Next, we turn our attention to the linear regression analysis (GLM-1) and their regression coefficients. In GLM-1, we implemented two task-related regressors, namely the subjective value of the current trial and the subjective value of the previous trial. The GLM was performed on each electrode contact and for each time point within the trial epoch separately. Let $v$ denote task-related variable. Here, we have two task related variables, the current and the previous subjective value. Let $\beta_{v,t}$ denote the regression vector consisting of regression coefficient of each contact associated with task variable $v$ (current subjective value or previous subjective value) at time $t$. $\beta_{v,t}$ therefore has a length of $N_{contacts}$. We then applied the $D$ to $\beta_{v,t}$ to denoise the regression vector

$$\beta_{v,t}^{pca} = D\beta_{v,t} \qquad (3)$$

where $\beta_{v,t}^{pca}$ is the denoised regression vector.

For each task variable $v$, we find the time $t_v^{max}$ that has the maximum L2 norm of $\beta_{v,t}^{pca}$ and define the corresponding regression vector $\beta_{v,t_v^{max}}^{pca}$ as $\beta_v^{max}$ where $\beta_v^{max}$ is the time-independent, de-noised regressor vector for task variable $v$. We then put $\beta_v^{max}$ from different $v$ (current and previous subjective value) together into a single two-dimensional

matrix $\beta^{max}$ where the columns are the $v$ and the rows are the electrode contacts. $\beta^{max}$ therefore has a size of $N_v \times N_{contacts}$ where $N_v = 2$ and $N_{contacts} = 166$. Finally, we obtain the orthogonal axes of the current subjective value and the previous subjective value by orthogonalizing $\beta^{max}$ with QR decomposition

$$\beta^{max} = QR \qquad (4)$$

where $Q$ is an orthogonal matrix and $R$ is an upper triangular matrix. The first two columns of $Q$ correspond to the orthogonalized regressor vectors $\beta_v^{\perp}$, which we refer to as the task-related axes of the current subjective value and the previous subjective value.

To study the representations of the current and previous subjective value in the OFC, we projected the condition-specific data matrix $X_c$ onto the orthogonal axes

$$P_{v,c} = \beta_v^{\perp T} X_c \qquad (5)$$

Where $P_{v,c}$ is the set of timeseries vectors over all $v$ and conditions. Here, we have two $v$, the current and previous subjective value, and four conditions. Therefore, in the two-dimensional space with the current and previous subjective value as the task-related axes, we have four trajectories, which we plotted in Fig. 9e–g.

## Reporting summary

Further information on research design is available in the Nature Portfolio Reporting Summary linked to this article.

## Data availability

Source data and statistics used to make each figure are provided in this paper. The high-gamma activity data generated in this study have been deposited in Open Science Framework (https://osf.io/f92yv/?view_only=95befcd791b9429692fc804a2876918a). The raw data analyzed in this study are available upon request made to the corresponding authors. Source data are provided in this paper.

## Code availability

The custom computer code used for the analyses reported in this study is available at Open Science Framework (https://osf.io/f92yv/?view_only=95befcd791b9429692fc804a2876918a).

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

## Acknowledgements

This work was supported by the National Science and Technology Council (NSTC) in Taiwan (Grants 108-2410-H-010 -012-MY3, 110-2410-H-A49A-504-MY3 to S.-W.W.) and by the Brain Research Center, National Yang Ming Chiao Tung University from The Featured Areas Research Center Program within the framework of the Higher Education Sprout Project by the Ministry of Education (MOE) in Taiwan. We acknowledge magnetic resonance imaging support from National Yang Ming Chiao Tung University, Taiwan, which is in part supported by the Ministry of Education plan for the top University.

## Author contributions

W.-Y.S.: conception and design of the work; acquisition, analysis, and interpretation of data; writing of the manuscript. H.-Y.Y.: acquisition of data and writing of the manuscript. C.-C.L.: acquisition of data and writing of the manuscript. C.-C.C.: acquisition of data and writing of the manuscript. C.C.: acquisition of data and writing of the manuscript. P.W.G.: conception and design of the work; analysis and interpretation of data; writing of the manuscript. S.-W.W.: conception and design of the work; analysis and interpretation of data; writing of the manuscript.

## Competing interests

The authors declare no competing interests.
