## [Peer Review File · Nature Communications]

Electrophysiological population dynamics reveal context dependencies during decision making in human frontal cortexREVIEWER COMMENTS

Reviewer #1 (Remarks to the Author):

In their article titled “Electrophysiological population dynamics reveal context dependencies during decision making in human frontal cortex”, Shih et al. investigated subjective value representation during a BDM task in rare electrophysiological recordings in humans. They found that the subjective value of the current trial is represented positively in the orbitofrontal cortex and that the subjective value of the past trial is represented negatively in the same brain region, but mainly on dissociated micro-sites. Overall, the results are very interesting, and the analyses are well conducted.

I have several concerns. Mainly about the results interpretation, the article structure and statistics, and the baseline correction.

1. I have concerns regarding the framing/interpretation of these results. While the authors frame their article around context-dependency, the task design is not well suited for context-dependency per se, as they do not test different contexts. By context, they mean the previous trial. This is confusing regarding the previous literature, which dissociates context-dependency, signal autocorrelation and adaptive coding. To me, their results are related to the latter, but their design does not really allow testing it directly, as the range of value is constant across blocks of trials. One alternative interpretation is that the BDM task induces a perceived cost at trial t and is then considered in trial $t+1$. Such interpretation could be supported by the positive representation of the previous value in the insula they observed (see Gueguen et al. 2021). My main question is then: how do authors dissociate those four different interpretations (context, autocorrelation, adaptive coding, cost) and why did they favor the context-dependency interpretation?

2. In terms of structure, the article could be much improved. For instance, in the introduction, there is a massive part of discussion of their results. The introduction should focus more on the previous literature.

3. Overall, no statistics are reported in the results, making the reviewing procedure difficult. There is no effect size or t-statistics, and solely mentions of p-values lower than 0.05 and stars on the figures. The results are written as if they were a legend for all figures, without reporting enough details on the conducted analyses. For instance:

- Page 8: “Across these OFC contacts, we found that high-gamma (80-150 Hz) activity represented both the current subjective value (the willingness-to-pay of the current trial) and the past subjective value

(the willingness-to-pay of the snack food item in the previous trial).". The term "represented" is already an interpretation. What are the statistics of this statement? What is the analysis?

- Page 9: "Among the significant OFC contacts, about half of them significantly represented the current subjective value, 29% of the contacts represented the previous subjective value, and 16% represented both the current and previous subjective value.". What is "about half of them"? Is it 50%? If so, what do the remaining 5% of sites code for? Again, this lack of precision downweighs the quality of the manuscript.

The full results section is written in this form and should be re-written.

4. The authors applied baseline correction:

- Which one exactly?

- How does this affect the results?

- Why do they observe a pre-stimulus correlation with value in low frequencies if such a correction has been applied?

- If the pre-stimulus value of each trial serves as a baseline for each trial, could the negative correlation with the previous value be an artifact from this correction? (The higher the previous value, the stronger the correction, thus triggering a negative correlation with the previous trial value only (t-1) and not t-2?).

5. I have concerns regarding figures 3A, B, and D. As we do not see the statistics, only the figures can speak for the data. Those three figures look strictly identical while they are generated from different GLMs; it is very unlikely that the results are not at least slightly affected by the used GLM. The authors should double-check whether they included the appropriate figures or not.

6. GLM2 results do not make sense. The author claim that they regress the residuals from the regression of signal against the current value and observe a significant effect of current value. This is mathematically not possible. What do they plot in figure 3B? The results from the regression of signal against the current value without the previous value as a second regressor?

7. Regarding behavior, is it common to observe a positive correlation between previous and current bids? Overall, can the behavior of the patients be considered as healthy?

8. The intertrial interval varies among 1s, 1.5s, and 2s. Is there any difference in the effect of the previous value between those timings? If so, could this be due to signal autocorrelations?

9. The time window on which the signal is investigated could be mentioned in the results, and not only in the methods.

10. References should be double-checked. For instance, Lopez-Persem et al., 2019 is not from Pessiglione and colleagues, as the authors claim in the introduction. Cavanagh et al. 2016 and Murray et al. 2014 are articles about auto-correlations, not context-dependencies. In the discussion, the authors claim that there is no study comparing the behavior from different value elicitation task but there is one: Lopez-Persem et al, Plos Comp Biol 2017.

11. I am not sure about what brings the trajectory analyses. Maybe the authors could better justify/explain it.

Reviewer #2 (Remarks to the Author):

Shih and colleagues investigated the neural representation of subjective value as measured by willingness to pay for food items in patients with epilepsy. Using stereo encephalography, they focus particularly on the orbitofrontal cortex (OFC) and report largely separate positive correlations of (high) gamma activity with current subjective value and negative correlations with preceding subjective value. Moreover, subjective value signals were similarly common in the hippocampus and the insula.

The paper is well written (apart from multiple typos) and makes a valuable contribution to the literature, particularly with a couple of further clarifications.

Major

1) Given that 16% of contacts showed common coding, the authors highlight the separate coding of present and preceding value and discuss that this separation may be specifically human. Indeed, non-human primate research repeatedly reported common coding of value and context even in single neurons. However, this highlights a potential difference between the current definition of context and that of other lines of research, which defined context explicitly rather than implicitly, simultaneously rather than consecutively or over different time horizons than the preceding trial. By extension, the present findings may be less generalizable than the authors conclude (e.g., last sentence of the abstract) and other forms of context may be represented more strongly than the presently investigated form. A related point worth considering in this regard is that neurons encoding subjective value in a positive or negative fashion are intermixed and often occur in similar proportions in OFC. Accordingly, human neuroimaging often uses multivariate analyses to isolate value signals, at least in central parts of OFC (whereas the univariate analyses more consistently identify value signals in medial parts of OFC). In any case, a more thorough discussion of, and engagement with, the literature would be desirable

2) You may call me old-fashioned, but I do not really see what the PCA and particularly the regression subspace analyses add over and above analyses like those illustrated in Figure 2B. For example, the conclusion that “context is a major source of organized patterns of activity in the OFC” can be drawn, also in a temporally resolved but more easily interpretable manner, with the earlier analyses. The meaning of principal components is stated to reflect current and preceding subjective value but it is not entirely clear to me what this statement is based on. Moreover, the wording appears to imply an interaction of current and preceding value but such an interaction has not been shown statistically anywhere, as far as I can see

3) Although the authors aimed to minimize motor confounds by temporally separating evaluation and response stages, participants could in principle have prepared their motor responses already in the evaluation stage, given that the response matrix was the same throughout. This could be mentioned briefly as limitation in the discussion

Intermediate and minor

1) The paper underrepresents the extent to which previous literature investigated and characterized context effects in value representations. In the individual domain, examples worth mentioning include:

a. Nieuwenhuis S et al., 2005, Neuroimage

b. Winston JS et al., 2014, JNeurosci

c. Burke CJ et al., 2016, JNeurosci

d. Pischedda D et al., 2020, JNeurosci

2) Given the positive relation between current and preceding subjective value, the question arises whether there was also a positive relation between the current subjective value (trial n) and that in trial n-2? This could be illustrated in a similar format as that of Figure 1C in the supplementary material

3) Page 33: it is not entirely clear how the PCA actually denoised the data. Assume the data were riddled by strong artefacts – some principal components would explain them well but these components would not be removed according to the described procedure

4) Were all the neural data analyzed from 1 s before onset of the food item to 1.5 s after onset of the food item (as mentioned in the main text – other passages mention 1.5 s before to 2 s after onset)? If not, please explain rationale. More importantly, what is the rationale for limiting the analysis at 1.5 s? May some value responses take longer?

5) Within the OFC, the study reports interesting differences in value representation along the medio-lateral axis, which appears most extensively covered by the locations of electrode contacts. Can you also say something about the anterior-posterior or at least about the dorsal-ventral axis?

6) Figures 2A and 4A show multiple contacts outside the brain. Please explain/adjust figures if possible

7) Please add similar plots as figures 5A and 5B also for the non-OFC regions (amygdala, hippocampus, striatum, insula, ACC&MCC, PCC, IPS) in the supplementary material

8) The specifications of TFCE statistics – e.g., $E=2$, $H=2$, should be explained upon first use

9) Did subjects with more contacts in a given region contribute more heavily to aggregate findings for that region?

10) Is it coincidence that the three regions with the strongest value signals (OFC, hippocampus and insula) were also the three regions with the highest number of contacts?

11) Typos:

a. current subject value -> current subjective value

b. Together, through -> Together,

c. spilt/spit -> split

d. the by the -> by the

e. spare -> sparse

f. fixation cross presenting -> fixation cross presented

g. Subjects can take -> Subjects could take

h. We did, however, observed -> We observed

i. TFCE where we -> TFCE which we

j. revered -> reversed

Reviewer #1 (Remarks to the Author):

In their article titled “Electrophysiological population dynamics reveal context dependencies during decision making in human frontal cortex”, Shih et al. investigated subjective value representation during a BDM task in rare electrophysiological recordings in humans. They found that the subjective value of the current trial is represented positively in the orbitofrontal cortex and that the subjective value of the past trial is represented negatively in the same brain region, but mainly on dissociated micro-sites. Overall, the results are very interesting, and the analyses are well conducted.

I have several concerns. Mainly about the results interpretation, the article structure and statistics, and the baseline correction.

1. I have concerns regarding the framing/interpretation of these results. While the authors frame their article around context-dependency, the task design is not well suited for context-dependency per se, as they do not test different contexts. By context, they mean the previous trial. This is confusing regarding the previous literature, which dissociates context-dependency, signal autocorrelation and adaptive coding. To me, their results are related to the latter, but their design does not really allow testing it directly, as the range of value is constant across blocks of trials. One alternative interpretation is that the BDM task induces a perceived cost at trial t and is then considered in trial $t+1$. Such interpretation could be supported by the positive representation of the previous value in the insula they observed (see Gueguen et al. 2021). My main question is then: how do authors dissociate those four different interpretations (context, autocorrelation, adaptive coding, cost) and why did they favor the context-dependency interpretation?

We appreciate the reviewer's thoughtful analysis and agree that there are many ways to describe the dependency that we examined. Doubtless, we all agree that bid values show a dependency on recent history. We first showed that in the BDM in the Khaw et al paper (Khaw et al., 2017). The excellent point that the reviewer raises is whether this dependency is best described as a ‘context’ or can be described as a form of adaptive coding, or with some other mechanistic label like ‘cost’, or simply an artefact of signal autocorrelation. In responding, we wanted to begin by clarifying our use of the word context, leaving open the possibility that there may be a better word and making clear that we would be happy to rephrase in a further revision. For us, context is meant to be a general category for all spatial and temporal influences that might impact, in this case, subjective value. Hence, we use the term in a statistical sense, much as studies of contrast gain in area V1 refer to the recent history of presented image contrasts as a context. To place the notion of context inside a very specific model, following the sensory tradition, if we think about divisive normalization models of context, this would be formally defined as anything that falls into the denominator. That would include anything in the exposure history or in the current spatial layout. We now try to make this clearer in the text and hope we have succeeded.

On adaptive coding: We acknowledge that the temporal term in all of the standard sensory models of this type is often described as the ‘adaptation’ term or the ‘adaptive coding’ term – one of the reviewer’s proposed alternative interpretations. Ohzawa and colleagues used the formulation ‘adaptation’ in their classic studies of V1 neurons in the mid-1980s where neuronal activity adapts to the mean contrast level of the visual stimulus (Ohzawa et al., 1982; Ohzawa et

al., 1985). This phenomenon is also often called contrast gain control. Because the spatial and temporal scales are the key factors for determining the mean contrast level, adaptation naturally is affected by both the spatial and temporal context of the environment (Carandini & Heeger, 2002). Following that usage of ‘adaptation’, many other labs who went on to describe center-surround organizations as relying on a similar mechanism introduced the notion that the spatial context terms could in parallel be called ‘spatial adaptation’. Indeed, in many of our and others’ publications we refer generally to context dependency as being made up of ‘spatial and temporal adaptation. For example, in Padoa-Schioppa (2009), what he referred to as range adaptation is in fact a particular form of “temporal context dependency”. So for us, and maybe some others in the field at least, we see context dependency as the super category, and spatial and temporal adaptation/context dependency as subcategories. We elected to refer to the effect we saw as context dependency in this more general sense. The context dependency we observed is a form of temporal adaptation/context dependency. In this we follow an existing usage and a literature in which our previous work is embedded, but again, we would be happy to employ another term if the reviewer feels strongly about it.

On signal autocorrelation: Turning next to the specific use of autocorrelation, what we believe to be the reviewer’s concern is the possibility that the previous-value findings (the impact of the last trial values) are simply due to the autocorrelation of the signals (in the high-gamma power). This is a legitimate concern and an excellent point which we did not sufficiently address. We have now addressed this by performing several new analyses, which are described below in the response to the reviewer’s comments #4 (baseline correction) and #8 (signal autocorrelation). We think that the best way to resolve this issue is by including *both* the previous subjective value and the signal from the previous trial in the same GLM – essentially the analysis proposed by the Nobelist Clive Granger to address this issue. In one analysis conducted in this manner (shown in response to comment #8), we performed a GLM analysis with the following regressors: the current subjective value, the previous subjective value, and the signal from the previous trial. In summary, we did not find the signal autocorrelation to significantly affect the previous subjective-value findings. As you will see below, additional analyses yield similar results indicating that this is not simply due to an inflexible autocorrelative mechanism.

On cost: Turning finally to cost, we now emphasize more clearly that only one trial would be realized at random at the end of the experiment and that subjects should treat each trial as equally likely to be *the* trial where the BDM procedure would be implemented. The Gueguen et al. paper, which the reviewer referenced, employed a task where the subjects received reward feedback after making each choice. Thus in that experiment, the chosen option was realized after each trial. This is unlike our task – we did not perform the auction after each trial and subjects received no feedback, and did not actually pay any cost, until the end of the experiment. Also recall that the actual paid cost from the subject lies randomly between 0 and the maximum amount that they bid. Subjects do not know the cost that they will pay until the end of the experiment. We think this an important point because a sequential dependency on costs actually paid is, we believe, more likely to be introduced when there is feedback, a revelation of what the costs actually are, after each trial. A range of studies show that with trial-by-trial revelation of the costs and benefits, people are more likely to show house money effect – an effect similar to perceived cost mentioned by the reviewer (Thaler & Johnson, 1990) – and are more likely to maximize expected value (Barron & Erev, 2003). This is the primary reason why many decision-

making experiments and neuroeconomic experiments that followed elect not to realize the chosen option after each trial (e.g., Tom et al., 2007). We hypothesize that it is less likely that trial t introduced a perceived cost that impacts trial $t+1$ primarily because subjects simply did not know what the cost associated with that previous trial and knew that there was a very low probability of the trial being selected for realization.

To address these important comments, however, we now include an expansion of the Introduction and Discussion. The Introduction now includes:

Page 3:

“Very broadly, context can be seen as a general category for describing the impact of the environment – most notably its spatial and temporal profiles – on brain activity and behavior. The process that leads to subsequent changes in behavior and brain activity is often referred to as adaptation and adaptive coding respectively²⁶⁻³¹.”

We also wish to point out in our original writing we emphasized our focus on temporal context dependency on pages 4-5 which we believe complements the above:

“First, and most importantly, we sought to determine whether human intracranial electrophysiological signals encoding rewards show a clear and ubiquitous context dependency, as has been observed in animals. To that end, we focused the inquiry on *temporal context dependency* and sought to gather evidence indicating whether or not the recent history of rewards influences the electrophysiological representation in reward-encoding areas of the human brain.”

That should, we hope, eliminate any uncertainty in our exposition. To address the larger questions about specific functions, cost, and autocorrelations, we have also added a paragraph to the Discussion where we reprise this point, and provide a larger ‘context’ (pun intended) for our work in the domains of spatial and temporal adaptation and with regard to autocorrelation. Specifically on pages 31-32 in Discussion:

“Although we found context-setting signals – the representations of previous subjective value – in the OFC and other brain regions, there are several alternative interpretations of the previous-value representations worth discussing. First, it is possible that the previous-value signals simply reflect the autocorrelation in the neural signals. To address this issue, we performed additional analyses and found that in general, signal autocorrelation did not significantly change the previous-value findings (Fig. 3FG; Supplementary Figs. 11-12). Second, it is possible that the BDM bidding task induces a perceived cost at trial t and is then considered in trial $t+1$. We think this might be less likely to occur as we did not realize the auction after each trial and because the actual paid cost in a BDM auction is determined randomly only at realization at the end of the experiment. The subjects were told that, at the end of the experiment, only one trial would be chosen at random and realized. Such design is an attempt to minimize the impact of previous decisions and is adopted by many decision-making experiments and neuroeconomic experiments⁸⁷. Indeed, when reward feedback is given after each trial, previous decision outcomes are more likely to incur a mental cost that would make people to want to break even (e.g., house money effect⁸⁸). People are also more likely to maximize expected value when reward feedback is given after each trial⁸⁹. Finally, we wish to acknowledge that context

dependency here can also be described as ‘adaptation’ or ‘adaptive coding’. We regard context dependency as the super category, and spatial and temporal adaptation as subcategories. We elected to refer to the effect we saw as context dependency in this more general sense. In the mid-1980s, Ohzawa and colleagues^{26,90} used the formulation ‘adaptation’ in their classic studies of V1 neurons where neuronal activity adapts to the mean contrast level of the visual stimulus. This phenomenon was termed contrast gain control. Because the spatial and temporal scales are the key factors for determining the mean contrast level, adaptation naturally is affected by both the spatial and temporal context of the environment⁹¹. Following that usage of ‘adaptation’, many other labs who went on to describe center-surround organizations as relying on a similar mechanism introduced the notion that the spatial context terms could in parallel be called ‘spatial adaptation’. Indeed, in our work and others context dependency is generally referred to as being made up of ‘spatial and temporal’ adaptation^{27,28,31}.”

2. In terms of structure, the article could be much improved. For instance, in the introduction, there is a massive part of discussion of their results. The introduction should focus more on the previous literature.

Thank you for the suggestion. We have now focused more on the previous literature, in particular, by engaging more on the context literature, and shortened the discussion of our results.

Specifically, we added on page 4:

“Most studies to date showed context-dependent neural responses either when participants experienced an outcome (e.g., monetary gain or loss) or were presented with reward-predicting cues^{30,53–56}. These context-dependent representations were observed in the standard subjective-value network including the ventromedial prefrontal cortex (vmPFC), striatum, and/or orbitofrontal cortex. However, unlike in macaques, it is unclear in humans whether and how the orbitofrontal cortex participates in context-dependent valuation during decision making^{29,57}. For example, at the time of choice, some brain regions in the subjective-value network assessed with fMRI would show context-dependent responses (ventral striatum) while others do not (vmPFC)²⁹.”

We also removed the following paragraph from the original manuscript:

“The study reported here was thus designed to allow us to examine several subareas of the frontal and orbital cortex at relatively high resolution, as well as providing data at areal levels of analysis throughout several nodes of the reward value-network; the amygdala, hippocampus, striatum, insula, cingulate cortex, and parietal cortex. Our work extends critical earlier work by Lopez-Persem and colleagues (Lopez-Persem et al., 2020) and Saez and colleagues (Saez et al., 2018) who were among the first to bridge the gap between human and animal research. Recording field potentials from human patients performing both rating and choice tasks, these authors provided the first electrophysiological confirmation that subjective value representations arise in the human brain in a manner very similar to what has been observed in the monkey.”

We moved our mentioning of the Lopez-Persem et al. and Saez et al. papers (Refs #58, 59) to the following paragraph on page 5:

“Here we report the use of stereo electroencephalography (sEEG) to record neural activity in human epileptic patients ($n=20$) performing an incentive compatible valuation task known to induce temporal context dependency at the behavioral level in humans⁴¹. **Building on recent human intracranial work in decision making**^{58,59}, our data show some of the first evidence for neurobiological temporal context-dependent value computations in humans.

We removed the part where we mentioned the trajectory analysis:

“Finally, we adapted advanced dimensionality-reduction methods (Mante et al., 2013) developed to analyze neuronal population activity in macaques to characterize the population responses in human frontal cortex. To our surprise, we found that human OFC activity traversed in a low-dimensional subspace with value and context as two orthogonal axes. More importantly, as a decision is being made, the temporal trajectory in this low-dimensional subspace began to distinguish the impact of the current and past information about reward value.”

3. Overall, no statistics are reported in the results, making the reviewing procedure difficult. There is no effect size or t-statistics, and solely mentions of p-values lower than 0.05 and stars on the figures. The results are written as if they were a legend for all figures, without reporting enough details on the conducted analyses. For instance:

Thank you for the suggestion. We have now included information about the t statistics in the main text (page 9) and also in the supplement (Supplementary Fig. 9). We also describe the procedure employed for our statistical tests and rewrote with more clarity and details in the Results section. Specifically, we added on page 9:

“After data preprocessing, we performed time-frequency analysis, separately for each trial, on the preprocessed local field potential (LFP) data in order to extract the timeseries data of oscillatory power associated with different frequency bands. The resulting timeseries data started from 1 s before the onset of the food stimulus to 1.5 s after the stimulus onset. The 1.5 s post-stimulus time window was approximately the average of the median response time across all subjects (1.5011 s; Supplementary Fig. 8). We recognize that if we set a longer time window, we could capture the dynamics of valuation more comprehensively in trials with longer response times. However, this would indicate, for shorter response-time trials, including time points after the subjects had already made a decision and entered into the response phase of the trial. Hence, choosing the time window at this length was an attempt to strike a balance between these opposing factors.

We subsequently performed General Linear Model (GLM) analyses on the power timeseries data with the subjective value of the current trial and the previous trial as two separate regressors. In particular, we focused on the high-gamma power (80-150 Hz) and referred to it as high-gamma activity to be consistent with the literature⁶¹. Across these OFC contacts, we found that high-gamma activity positively correlated with the current subjective value (maximum t

statistic=7.0229 at 710 ms after stimulus onset) but negatively correlated with the previous subjective value (minimum t statistic=-4.8888 at 1020 ms after stimulus onset) (Fig. 2B; $p<0.05$, familywise error corrected for multiple testing across time points). A version of these results in the form of the t statistic of the regression coefficients can be seen in the Supplement (Supplementary Fig. 9). Statistical significance was established based on a permutation test with threshold-free cluster enhancement (TFCE) as the test statistic^{64,65}. The TFCE statistic is used to describe signals that exhibit some spatial or temporal continuity without having to arbitrarily set a threshold for defining clusters of signals. For a given regressor of interest, we computed the TFCE statistic based on the z -statistic timeseries of its corresponding regression coefficients and for each time point separately. Through random permutations we estimated the null distribution of the maximum TFCE statistic, which was then used for statistical inference (see *Group-level permutation test* in Methods for details).”

- Page 8: “Across these OFC contacts, we found that high-gamma (80-150 Hz) activity represented both the current subjective value (the willingness-to-pay of the current trial) and the past subjective value (the willingness-to-pay of the snack food item in the previous trial).”. The term “represented” is already an interpretation. What are the statistics of this statement? What is the analysis?

We rewrote the first paragraph the reviewer pointed out to include a description of the analysis and results. Specifically, to answer the reviewer’s question, we now add on page 9:

“We subsequently performed General Linear Model (GLM) analyses on the power timeseries data with the subjective value of the current trial and the previous trial as two separate regressors. In particular, we focused on the high-gamma power (80-150 Hz) and referred to it as high-gamma activity to be consistent with the literature⁶¹. Across these OFC contacts, we found that high-gamma activity positively correlated with the current subjective value (maximum t statistic=7.0229 at 710 ms after stimulus onset) but negatively correlated with the previous subjective value (minimum t statistic=-4.8888 at 1020 ms after stimulus onset) (Fig. 2B; $p<0.05$, familywise error corrected for multiple testing across time points). A version of these results in the form of the t statistic of the regression coefficients can be seen in the Supplement (Supplementary Fig. 9). Statistical significance was established based on a permutation test with threshold-free cluster enhancement (TFCE) as the test statistic^{64,65}. The TFCE statistic is used to describe signals that exhibit some spatial or temporal continuity without having to arbitrarily set a threshold for defining clusters of signals. For a given regressor of interest, we computed the TFCE statistic based on the z -statistic timeseries of its corresponding regression coefficients and for each time point separately. Through random permutations we estimated the null distribution of the maximum TFCE statistic, which was then used for statistical inference (see *Group-level permutation test* in Methods for details).”

- Page 9: “Among the significant OFC contacts, about half of them significantly represented the current subjective value, 29% of the contacts represented the previous subjective value, and 16% represented both the current and previous subjective value.”. What is “about half of them”? Is it 50%? If so, what do the remaining 5% of sites code for? Again, this lack of precision downweighs the quality of the manuscript.

The full results section is written in this form and should be re-written.

We rewrote the Results section as the reviewer suggested to improve precision in describing our results.

To answer the reviewer's specific questions regarding

Q: What is "about half of them"? Is it 50%?

A: It is 52% to be precise

Q: If so, what do the remaining 5% of sites code for?

A: We apologize for not being precise in our statement. 52% represented only the current value, 31% represented only previous value, and the remaining 17% represented both the current and previous value.

This is now clarified on page 10 in the revised manuscript:

"Among the significant OFC contacts, 52% significantly represented only the current subjective value, 31% of the contacts represented only the previous subjective value, and 17% represented both the current and previous subjective value."

4. The authors applied baseline correction:

- Which one exactly?

We applied baseline correction in two places. First, in the final stage of data preprocessing, we applied baseline correction on the preprocessed LFP data. The baseline is the average amplitude (voltage) of the 1.5-s time window before the onset of the food stimulus. We clarified this on page 40:

"The epoch for each trial (trial epoch) started 1.5 s before the onset of the food stimulus and ended 2 s after stimulus onset, with a pre-stimulus baseline correction where the average activity of the 1.5-s time window before the onset of the food stimulus was treated as the baseline."

Second, after performing time-frequency analysis, for each frequency bin separately, we computed the mean power (averaged across time within the 1-s time window before the onset of the food stimulus) and used it as the baseline. We then subtracted baseline from the timeseries data of spectral power. We clarified this on page 40:

"For each frequency bin separately, we computed the mean power (averaged across time within the 1-s time window before the onset of the food stimulus) and used it as the baseline. We then subtracted baseline from the timeseries data of spectral power."

- How does this affect the results?

Thank you for raising this important question. As the reviewer pointed out in the later comments, the negative correlation with previous value found in high-gamma activity (Fig. 3) could be an

artifact of our baseline correction. In particular, if the baseline activity – which is the average power of the pre-stimulus period – positively correlated with the previous subjective value, then subtracting it from the timeseries data could introduce a negative correlation with the previous subjective value during the post-stimulus period. In order to address this question, we had to first examine whether the baseline activity – average power of the pre-stimulus period – correlated with the previous subjective value. We found that, across different frequency bands, the average power of the pre-stimulus period did not significantly correlate with the previous subjective value.

We performed linear regression analysis where we regress baseline activity – the average power of pre-stimulus period – against previous subjective value. Here we plot the regression coefficient of previous subjective value at different frequency bands (High gamma, Gamma, Beta, Alpha, Theta). We did not find the regression coefficients to be statistically different from 0. The t statistic and p-value associated with each frequency band are included in the figure. We now include this figure in the Supplement (Supplementary Fig. 11).

- Why do they observe a pre-stimulus correlation with value in low frequencies if such a correction has been applied?

From the statistical standpoint, this suggests that at lower frequencies the time-wise deviation of the power timeseries from the mean during the pre-stimulus period carried task-related information. In this case, such task-related information is the subjective value.

From a conceptual standpoint, this might be related to the variability of the subjects' bids, as we described in the original text below on page 19:

“Noticeably, the low-frequency activity significantly represented a bias on the current subjective value before stimulus onset—before information about the current food item was revealed. We found that in the alpha band, these results were associated with two behavioral patterns: the variability of the bids and the correlation between the subjects' stated current subjective value and the subjects' stated previous subjective value. As might be expected, we found that, in part, such pre-stimulus representations were driven by the subjects who showed less variability in their bids and whose bids were more affected by the bid in the previous trial (Supplementary Figs. 14 and 15). In other words, the more the subjects relied on the previous bid, the greater the likelihood of significant pre-stimulus representations (a form of bias).”

- If the pre-stimulus value of each trial serves as a baseline for each trial, could the negative correlation with the previous value be an artifact from this correction? (The higher the previous value, the stronger the correction, thus triggering a negative correlation with the previous trial value only (t-1) and not t-2?).

To directly address the reviewer's question, we looked at the baseline activity itself and examined whether it correlated with the previous subjective value. Here the baseline is defined as the average power of the pre-stimulus period. The results are shown below. For each trial separately, we computed the average power during the pre-stimulus period (averaged across time points during the pre-stimulus period). We then used this as data and regressed them against the previous subjective value (see graph below). We found that the pre-stimulus activity did not correlate with previous subjective value. If pre-stimulus activity did not correlate previous subjective value, then it would be less likely to explain the previous-value findings as an artifact of the baseline correction.

We performed linear regression analysis where we regress baseline activity – the average power of pre-stimulus period – against previous subjective value. Here we plot the regression coefficient of previous subjective value at different frequency bands (High gamma, Gamma, Beta, Alpha, Theta). We did not find the regression coefficients to be statistically different from 0. The t statistic and p-value associated with each frequency band are included in the figure. We now include this figure in the Supplement (Supplementary Fig. 11).

5. I have concerns regarding figures 3A, B, and D. As we do not see the statistics, only the figures can speak for the data. Those three figures look strictly identical while they are generated from different GLMs; it is very unlikely that the results are not at least slightly affected by the used GLM. The authors should double-check whether they included the appropriate figures or not.

Thank you for really looking carefully on the figure. We double checked the analysis to make sure this was not due to some mistakes. When we first obtained the results, we were also shocked by their similarities. We therefore checked plenty of times, reran the analyses, and obtained the same results. We now add the *t*-statistic version of Fig. 3 in the supplement (Supplementary Fig. 13) which does reveal a small difference, as expected. We show Supplementary Fig. 13 below.

6. GLM2 results do not make sense. The author claim that they regress the residuals from the regression of signal against the current value and observe a significant effect of current value. This is mathematically not possible. What do they plot in figure 3B? The results from the regression of signal against the current value without the previous value as a second regressor?

We apologize for the confusion we accidentally misrepresented what we had done. GLM-2 consisted of two separate regressions performed in two steps. In the first regression, we regressed brain activity (high-gamma power or gamma power) against only the current value (no previous value as regressor). In the second regression, we used the residuals from the first regression as data and regressed them against only the previous value (no current value as regressor). In Figure 3B, we plot the regression coefficient of the current value from the first regression, and the coefficient of the previous value from the second regression. In other words, the first regression only contained the current value and the constant as the regressors, and the second regression only contained the previous value and constant as regressors. The data in the second regression were the residuals from the first regression. We also did the analysis in reversed order and found similar results (Supplementary Figure 10). Below we show Supplementary Fig. 10.

We updated our descriptions in the manuscript on pages 12-13 to eliminate this error: “GLM-2 consisted of two separate regressions performed in two steps. In the first regression, we regressed high-gamma activity against only the current subjective value. We did not include the previous subjective value as regressor. Then, in the second regression, we used the residuals from the first regression as data and regressed them against only the previous subjective value. In this second regression, we did not include the current subjective value as regressor. We then plotted the regression coefficient of the current subjective value estimated from the first regression, and the regression coefficient of the previous subjective value estimated from the second regression (Fig. 3B). The results still indicated that the high-gamma activity in the OFC positively correlated with the current subjective value and negatively correlated with the previous subjective value (Fig. 3B)—consistent with the original model (GLM-1 in Fig. 3A). We also performed this two-step regression analysis in the reversed direction (regress high-gamma activity against previous subjective value in the first regression and regress the residuals from the

first regression against the current subjective value in the second regression) and the results were identical (See Supplementary Fig. 10).”

7. Regarding behavior, is it common to observe a positive correlation between previous and current bids? Overall, can the behavior of the patients be considered as healthy?

First, we note that Khaw and colleagues’ paper showed just such a correlation in healthy subjects. Recent work by Christine Constantanople’s lab shows a similar result in healthy rodents. However, to clarify this point we ran an additional experiment. We now present new behavioral data collected from 35 healthy subjects performing the same BDM task with identical food items. The results are shown below. Overall, we found that most subjects (25 out of 35 subjects) showed a positive correlation between current and previous bids. Among all subjects, five subjects’ current bids were significantly and positively correlated with previous bids. Only one subject’s current bids were significantly and negatively correlated with previous bids. These results are consistent with the patients’ results. We now include the results in Supplementary Fig. 2.

Specifically, we mentioned in the Results section on page 8:

“To further examine whether such temporal context dependency can be considered normal, we ran the same task on 35 healthy subjects from the normal population and found that most subjects (25 out of 35) showed the same temporal context dependency, with the current bid positively correlating with the bid on the previous trial (Supplementary Fig. 2; see also ref⁴¹).”

And in the Methods section when we described Participants (page 37):

“To learn more about the patients’ decision-making behavior in relation to the normal population, we collected behavioral data from 35 healthy subjects in the normal population not treated for epilepsy (18 males, aged 20-27 years; average: 22 years) performing the same task. This part of the study was approved by National Yang Ming Chiao Tung University Institutional Review Board. Informed consent was obtained from each participant before participation.”

8. The intertrial interval varies among 1s, 1.5s, and 2s. Is there any difference in the effect of the previous value between those timings? If so, could this be due to signal autocorrelations?

Thank you for the suggestion. We ran the GLM analysis separately for different ITIs. The results are shown below. Overall, the results are very similar between different intertrial intervals. High-gamma activity significantly and negatively correlated with previous subjective value in all three different intertrial intervals. Qualitatively, we did notice that when the intertrial interval was 1s (shortest), the temporal extent of the previous-value representation was smaller than longer intertrial intervals at 1.5s and 2s. While the mechanism underlying this pattern is unclear, it would argue against signal autocorrelations, since autocorrelation would predict stronger previous-value effect at shorter intertrial intervals.

We add this figure in the revised manuscript (Fig. 3G).

We further examined the effect of signal autocorrelation with two additional analyses. First, we estimated signal autocorrelation by computing the Pearson correlation of the high-gamma power of trial t with the high-gamma power of trial $t-1$ in a timewise fashion. The result is shown in the figure below, which is added in the Supplement (Supplementary Fig. 12). In general, we found the signal autocorrelation – correlation in high-gamma power between adjacent trials – to be weak. Most of the correlation was 0 and at most -0.02 in some time points. Importantly, the correlation at 500 to 1000 ms after stimulus onset – the time period where we observed significant negative correlation with the previous subjective value – was 0. This suggests that the previous-value finding was not driven by the signal autocorrelation.

Second, we directly add the high-gamma activity of the previous trial as a regressor in addition to the current and previous value regressors in the GLM analysis. This is the strongest test for the effect of previous-value on high-gamma activity because the model now includes signals from the previous trial to compete with the previous subjective value. The result is shown below. We found that the negative correlation with the previous subjective value (in green) still holds at 500 to 1000 ms after stimulus onset. In addition, we found that during this time period (500-1000 ms after stimulus onset), the power from the previous trial did not significantly correlate with the power on the current trial. This further ruled out signal autocorrelation as a potential explanation of the previous-value representations.

We add this figure in the revised manuscript (Fig. 3F).

Accordingly, we now mention these important new findings in the Results section (page 15):

“Fifth, we examined whether the negative correlation with the previous subjective value could arise from signal autocorrelation, the deterministic correlation in spectral power between two successive trials. To address this issue we modified GLM-1 by adding the power on the previous trial as a regressor in addition to the current and previous subjective value regressors. That is, for each time point separately, the spectral power of the same time point from the previous trial was used as the regressor. We found that the negative correlation with previous subjective value remained (Fig. 3F). Sixth, we examined whether the results on the previous subjective value could be affected by the length of the inter-trial interval (ITI) that we randomly varied across trials (1 s, 1.5 s, and 2 s). If signal autocorrelation did have an effect on the previous-value findings, we expect to observe weaker results on previous subjective value with shorter ITIs. We implemented GLM-1 separately for each possible ITI and found that the length of ITI did not affect results on the previous subjective value (Fig. 3G). The negative correlation with previous subjective value was significant across different ITIs. Further analyses on issues related to signal autocorrelation can be seen in the Supplement (Supplementary Figs. 11-12).”

We also included in the supplement the *t*-statistic version of Fig. 3 in the main text. We mentioned it on page 15: “The *t*-statistic version of these results can be seen in the Supplement (Supplementary Fig. 13).”

Supplementary Fig. 13:

9. The time window on which the signal is investigated could be mentioned in the results, and not only in the methods.

Thank you. We now mention it in the first paragraph describing the high-gamma results in the Results section on page 9:

“The resulting timeseries data started from 1 s before the onset of the food stimulus to 1.5 s after the stimulus onset.”

10. References should be double-checked. For instance, Lopez-Persem et al., 2019 is not from Pessiglione and colleagues, as the authors claim in the introduction. Cavanagh et al. 2016 and Murray et al. 2014 are articles about auto-correlations, not context-dependencies. In the discussion, the authors claim that there is no study comparing the behavior from different value elicitation task but there is one: Lopez-Persem et al, Plos Comp Biol 2017.

Thank you for pointing this out. We corrected these errors in the Introduction on page 5 (Ref 59 is Lopez-Persem et al. 2020):

“Building on recent human intracranial work in decision making^{58,59}, our data show some of the first evidence for neurobiological temporal context-dependent value computations in humans.”

We added Lopez-Persem et al. 2017 (Ref 84) in the Discussion and rewrote part of the paragraph on pages 29-30:

“Our current understanding of the subjective valuation of rewards in humans is based on three different methods for eliciting subjective value—choice tasks, the Becker-DeGroot-Marschak auction task where subjects indicate willingness-to-pay, and liking-rating tasks. At the behavioral level, earlier work highlighted inconsistencies between BDM task and choice task⁸⁰⁻⁸³. More recently, however, Lopez-Persem and colleagues⁸⁴ did find that different elicitations of subjective value (choice, rating, and effort tasks) tend to produce consistent results. However, at the neural level, there has been little discussion to date on how these different elicitation methods might differentially impact the subjective-value signals measured physiologically.”

11. I am not sure about what brings the trajectory analyses. Maybe the authors could better justify/explain it.

The trajectory analyses, by and large, revealed that the current and previous subjective value explained the most data variance across the OFC electrodes. We found that the first two principal components corresponded to the previous subjective value (first PC) and the current subjective value (second PC). We now include a paragraph at the beginning of the trajectory analyses to provide our reasons and justification on pages 23-24:

“In these analyses, we approached the OFC activity from different electrode contacts as a single population consisting of high-dimensional data (each contact serves as a dimension). Through dimensionality reduction, these analyses allowed us to characterize the dynamics of the high-

dimensional data as temporal trajectories in some low-dimensional space. An important question we asked is whether and how the axes forming the low-dimensional space relate to our task variables, namely the subjective value and temporal context. These analyses provided two major compliments to the GLM analyses. First, they serve to validate whether subjective value and temporal context dominated the electrophysiological signals in OFC as a population as is implied by our regression analyses and by the structure of the task we employed. Second, they provided alternative visualizations for how subjective value and temporal context impacted OFC population activity.”

Reviewer #2 (Remarks to the Author):

Shih and colleagues investigated the neural representation of subjective value as measured by willingness to pay for food items in patients with epilepsy. Using stereo encephalography, they focus particularly on the orbitofrontal cortex (OFC) and report largely separate positive correlations of (high) gamma activity with current subjective value and negative correlations with preceding subjective value. Moreover, subjective value signals were similarly common in the hippocampus and the insula.

The paper is well written (apart from multiple typos) and makes a valuable contribution to the literature, particularly with a couple of further clarifications.

Major

1) Given that 16% of contacts showed common coding, the authors highlight the separate coding of present and preceding value and discuss that this separation may be specifically human. Indeed, non-human primate research repeatedly reported common coding of value and context even in single neurons. However, this highlights a potential difference between the current definition of context and that of other lines of research, which defined context explicitly rather than implicitly, simultaneously rather than consecutively or over different time horizons than the preceding trial. By extension, the present findings may be less generalizable than the authors conclude (e.g., last sentence of the abstract) and other forms of context may be represented more strongly than the presently investigated form. A related point worth considering in this regard is that neurons encoding subjective value in a positive or negative fashion are intermixed and often occur in similar proportions in OFC. Accordingly, human neuroimaging often uses multivariate analyses to isolate value signals, at least in central parts of OFC (whereas the univariate analyses more consistently identify value signals in medial parts of OFC). In any case, a more thorough discussion of, and engagement with, the literature would be desirable

Thank you for this great suggestion. We agree that context can mean different things, and that we should provide more discussion on the literature. We also rewritten the Abstract, toning down the generalizability claim in the last sentence. More detail can be found on this in the response to Reviewer#1

In the Discussion, we add on pages 31-32:

“Finally, we wish to acknowledge that context dependency here can also be described as ‘adaptation’ or ‘adaptive coding’. We regard context dependency as the super category, and spatial and temporal adaptation as subcategories. We elected to refer to the effect we saw as context dependency in this more general sense. In the mid-1980s, Ohzawa and colleagues^{26,90} used the formulation ‘adaptation’ in their classic studies of V1 neurons where neuronal activity adapts to the mean contrast level of the visual stimulus. This phenomenon was termed contrast gain control. Because the spatial and temporal scales are the key factors for determining the mean contrast level, adaptation naturally is affected by both the spatial and temporal context of the environment⁹¹. Following that usage of ‘adaptation’, many other labs who went on to describe center-surround organizations as relying on a similar mechanism introduced the notion that the spatial context terms could in parallel be called ‘spatial adaptation’. Indeed, in our work

and others context dependency is generally referred to as being made up of ‘spatial and temporal’ adaptation^{27,28,31}.”

To discuss the intermixing of positive and negative encoding of subjective value, we add on page 33:

“A related point is the intermixing of positive and negative encoding of subjective value. Our single-contact analysis showed that, for both the subjective value and temporal context, OFC contacts showed either positive or negative correlation with these variables. This is in part consistent with monkey electrophysiology studies showing intermixed encoding of subjective value in the OFC – some neurons positively correlated with subjective value, while others showed negative correlation⁶. As these positive and negative encoding neurons are similar in number, multivariate decoding analyses that take these neurons or populations as data should be able to readout these intermixed subjective-value signals. Indeed, human fMRI studies using multivoxel decoding analysis found subjective value signals in the central OFC⁴⁸ and context-dependent responses to reward outcomes in the ventromedial prefrontal cortex^{30,56}. Our single-contact results further suggested the possibility of decoding subjective value and context-setting signals in human fMRI with multivoxel decoding approach.”

2) You may call me old-fashioned, but I do not really see what the PCA and particularly the regression subspace analyses add over and above analyses like those illustrated in Figure 2B. For example, the conclusion that “context is a major source of organized patterns of activity in the OFC” can be drawn, also in a temporally resolved but more easily interpretable manner, with the earlier analyses. The meaning of principal components is stated to reflect current and preceding subjective value but it is not entirely clear to me what this statement is based on. Moreover, the wording appears to imply an interaction of current and preceding value but such an interaction has not been shown statistically anywhere, as far as I can see

Thank you for this. The truth is that we have generally railed against the contemporary overuse of PCA to restate the obvious. That said, as we have presented this result to younger audiences we have found that we are often asked to present a PCA. Like the reviewer, we view these analyses as (at best) a compliment the GLM analyses (at worst a restatement of those results). In any case, bowing to fashion, we do think that these analyses will be helpful for some younger readers. To further clarify and motivate these analyses, we now add the following in Results on page 23-24:

“In these analyses, we approached the OFC activity from different electrode contacts as a single population consisting of high-dimensional data (each contact serves as a dimension). Through dimensionality reduction, these analyses allowed us to characterize the dynamics of the high-dimensional data as temporal trajectories in some low-dimensional space. An important question we asked is whether and how the axes forming the low-dimensional space relate to our task variables, namely the subjective value and temporal context. These analyses provided two major compliments to the GLM analyses. First, they serve to validate whether subjective value and temporal context dominated the electrophysiological signals in OFC as a population as is implied by our regression analyses and by the structure of the task we employed. Second, they provided

alternative visualizations for how subjective value and temporal context impacted OFC population activity.”

With regard to the reviewer’s question on how principal components reflect current and previous subjective value, please look at Fig. 9C and 9D. In Fig. 9C, we plot the population activity projected onto PC-1. As can be seen in the graph, the temporal trajectories between high (yellow and cyan) versus low previous value (purple and magenta) began to separate after stimulus onset, suggesting that PC-1 reflects the previous value. In Fig. 9D, we plot the population activity projected onto PC-2. Here, we see that the trajectories between high (cyan and magenta) and low (purple and yellow) current subjective value began to separate within 500 ms after stimulus onset.

These are described on page 25-26:

“These directions revealed that the first principal component appears to capture the temporal context—subjective value observed in the previous trial (Fig. 9C), separating high previous subjective value (yellow and cyan) from low previous subjective value (purple and magenta). The degree of separation appears to depend on the current subjective value, with stronger separation between high and low previous subjective value when the current subjective value was high (cyan and magenta). The onset of this stronger separation emerged early, right after the stimulus onset (0 s mark). The second principal component appears to capture information in the population about the magnitude of the subjective value observed in the current trial (Fig. 9D), separating high current subjective value (cyan and magenta) from low current subjective value (yellow and purple). The emergence of this separation also appeared to be early, approximately 200 ms after stimulus onset.”

With regard to the interaction, we wish to emphasize that our intention was not to state that there is an interaction. We therefore remove any notion that would suggest an interaction between subjective value and temporal context.

3) Although the authors aimed to minimize motor confounds by temporally separating evaluation and response stages, participants could in principle have prepared their motor responses already in the evaluation stage, given that the response matrix was the same throughout. This could be mentioned briefly as limitation in the discussion

Thank you for the suggestion. We now include this when discussing the limitation in Discussion (page 34):

“One limitation of our task design is motor-related confounds. Although we aimed to minimize motor confounds by temporally separating the evaluation and the response stages, the subjects could in principle have prepared their motor responses during the evaluation stage given that the response matrix was the same throughout the experiment. Future investigations could address this issue by not specifying the motor-response mapping until the response stage. For example, one can design a visual analog scale for bid value and could randomly, from trial to trial, vary the presentation of bid value (left-to-right indicates low-to-high value in half of the trials and high-to-low value in the other half).”

Intermediate and minor

1) The paper underrepresents the extent to which previous literature investigated and characterized context effects in value representations. In the individual domain, examples worth mentioning include:

- a. Nieuwenhuis S et al., 2005, Neuroimage
- b. Winston JS et al., 2014, JNeurosci
- c. Burke CJ et al., 2016, JNeurosci
- d. Pischedda D et al., 2020, JNeurosci

Thank you very much for pointing out these papers. We now add these papers in the Introduction on page 4:

“Most studies to date showed context-dependent neural responses either when participants experienced an outcome (e.g., monetary gain or loss) or were presented with reward-predicting cues^{30,53–56}. These context-dependent representations were observed in the standard subjective-value network including the ventromedial prefrontal cortex (vmPFC), striatum, and/or orbitofrontal cortex. However, unlike in macaques, it is unclear in humans whether and how the orbitofrontal cortex participates in context-dependent valuation during decision making^{29,57}. For example, at the time of choice, some brain regions in the subjective-value network assessed with fMRI would show context-dependent responses (ventral striatum) while others do not (vmPFC)²⁹.”

2) Given the positive relation between current and preceding subjective value, the question arises whether there was also a positive relation between the current subjective value (trial n) and that in trial n-2? This could be illustrated in a similar format as that of Figure 1C in the supplementary material

Thank you. We now add the n-2 regression result in the supplement and noted it in the main text on page 8:

“Further analysis revealed that the bid in the trial presented two-trials back did not have a significant effect on the current bid (Supplementary Fig. 1).”

3) Page 33: it is not entirely clear how the PCA actually denoised the data. Assume the data were riddled by strong artefacts – some principal components would explain them well but these components would not be removed according to the described procedure

Sorry for the lack of clarity. Yes, the reviewer’s conjecture is correct. We did not aim to denoise the data with PCA. Rather, the goal was to identify and remove eye-movement artifacts. To do that, we proceeded in two steps. First, we performed PCA on the data and – by keeping the PCs that explained 95% of data variance – we effectively reduced the dimensions of the data. Second, using data based on PCs that explained 95% of data variance, we performed ICA. We then identified ICs that shared similar temporal profiles with the eye-movement ICs based on scalp EEG data. We note that because of the first step, the number of ICs identified in the second step would be greatly reduced compared with directly performing ICA on the raw data. And since eye-movement activities are typically very strong artifacts, this approach in principle would not compromise the ability to identify these ocular artifacts. Rather, it saved us significant computation time.

We rewrote the relevant paragraph on page 40:

“Fourth, in order to remove eye-movement-related activity from the sEEG data, we proceeded in two steps, which were separately applied to both the scalp EEG and sEEG data. First, we performed PCA on the data and – by keeping the PCs that explained 95% of data variance – we effectively reduced the dimensions of the data. Second, we performed ICA using data based on the PCs that explained 95% of data variance. Eye-related activities were first identified by inspecting the independent components (ICs) of the scalp EEG data. Once an IC with ocular

artifacts was identified, we checked whether there was a corresponding IC in the sEEG data. We note that because of the first step, the number of ICs identified in the second step would be greatly reduced compared with directly performing ICA on the raw sEEG data. And since eye-movement artifacts are mostly likely very strong artifacts, this approach in principle would not compromise the ability to identify these ocular artifacts. Rather, it saved us significant computation time.”

4) Were all the neural data analyzed from 1 s before onset of the food item to 1.5 s after onset of the food item (as mentioned in the main text – other passages mention 1.5 s before to 2 s after onset)? If not, please explain rationale. More importantly, what is the rationale for limiting the analysis at 1.5 s? May some value responses take longer?

Yes, we decided to analyze the neural activity (the power) with the same time window across all subjects, which is 1 s before stimulus presentation and 1.5 s after stimulus presentation. In our data preprocessing, the EEG epochs were slightly longer (1.5 s before to 2 s after onset) so that we can apply time-frequency analysis upon them without zero-padding. We clarified this in Methods on page 40:

“We note that in our data preprocessing, the EEG epochs were slightly longer (1.5 s before to 2 s after onset) so that we can apply time-frequency analysis upon them without zero-padding.”

Our rationale for choosing the 1.5 s post-stimulus time window: 1.5 s was approximately the average of the median response time across all subjects (1.5011 s). We recognize that if we set a longer time window, we could capture the dynamics of valuation more comprehensively in trials with longer response time. However, this would indicate, for shorter response-time trials, including time points after the subjects had already made a decision and entered into the response phase of the trial. Hence, choosing the time window at this length was an attempt to strike a balance between these opposing factors.

We clarified our rationale on page 9 in Results:

“After data preprocessing, we performed time-frequency analysis, separately for each trial, on the preprocessed local field potential (LFP) data in order to extract the timeseries data of oscillatory power associated with different frequency bands. The resulting timeseries data started from 1 s before the onset of the food stimulus to 1.5 s after the stimulus onset. The 1.5 s post-stimulus time window was approximately the average of the median response time across all subjects (1.5011 s; Supplementary Fig. 8). We recognize that if we set a longer time window, we could capture the dynamics of valuation more comprehensively in trials with longer response times. However, this would indicate, for shorter response-time trials, include time points after the subjects had already made a decision and entered into the response phase of the trial. Hence, choosing the time window at this length was an attempt to strike a balance between these opposing factors.”

We also include the distribution of median response time across all subjects in the supplement (Supplementary Fig. 8):

5) Within the OFC, the study reports interesting differences in value representation along the medio-lateral axis, which appears most extensively covered by the locations of electrode contacts. Can you also say something about the anterior-posterior or at least about the dorsal-ventral axis?

Yes. We now include the anterior-posterior results as Figure 5 in the revised manuscript. The Figure is also shown below.

Figure 5. Subjective-value representations in the anterior and posterior OFC. **A.** Electrode contacts in the anterior (left) and posterior OFC (right). **B.** High-gamma activity. Average time course of regression coefficients for the current subjective value (in blue) and previous subjective value (in green) in the medial, central, and lateral OFC. * indicates $p < 0.05$ (familywise error corrected) using permutation test based on the threshold-free cluster enhancement (TFCE) statistic. Colored (blue or green) horizontal lines indicate the time points with $p < 0.05$ (familywise error corrected). **C.** Subjective-value representations in individual OFC contacts. Conventions are the same as described in Fig. 2C.

6) Figures 2A and 4A show multiple contacts outside the brain. Please explain/adjust figures if possible

The background brain image was based on the MNI template. It shows the original surface, which is gray/white matter boundary. As a result, some contacts can appear to be outside the brain where in fact they are not. We identified the contacts that appeared to be outside of the brain and here are their coordinates in the MNI space (x-53, y42, z-11), (x-51, y33, z2), (x-49, y31, z0), (x54, y28, z2). Below we show these coordinates in the volumetric image of the MNI template. Information about the MNI coordinates of all electrode contacts mentioned in the main texts are in the supplement (Supplementary Tables 1-8).

(x-53, y42, z-11). This particular contact is borderline. But according to the Harvard-Oxford probabilistic atlas, it has a 25% of being part of the cortex. So we included it in the analysis.

(x-51, y33, z2)

(x-49, y31, z0)

(x54, y28, z2)

We now include a description of the surface template in Methods on pages 41-42:

“The MNI template that the electrode contacts were overlaid on was based on the surface that depicts the gray/white matter boundary. As a result, some contacts can appear to be outside of the brain where in fact they are not. We used the Harvard-Oxford probabilistic atlas to verify that these contacts are in fact inside the brain. Information about the MNI coordinates of all the electrode contacts mentioned in the paper are in the Supplement (Supplementary Tables 1-8).”

7) Please add similar plots as figures 5A and 5B also for the non-OFC regions (amygdala, hippocampus, striatum, insula, ACC&MCC, PCC, IPS) in the supplementary material

We now add them in the supplement and mentioned them in the main text on page 22:

“The cross-frequency representations of subjective value associated with these brain regions, like the ones shown in Fig. 6A for the OFC, can be seen in the Supplement (Supplementary Figs. 16-17).”

8) The specifications of TFCE statistics – e.g., $E=2$, $H=2$, should be explained upon first use

We now clarify the E and H parameters on page 43:

“The t statistic was transformed to the z statistic, and using the z statistic we computed the TFCE statistic ($E=2$, $H=2$) for each time point. The TFCE statistic summarizes the strength of spatially or temporally extended signals, i.e. spatial or temporal clusters. A major advantage of using the TFCE statistic is that we do not need to specify a cluster-forming threshold in order to identify spatial or temporal clusters of activation. The TFCE statistic is defined as the sum of the ‘scores’ of all supporting sections underneath the z statistic of a particular time point (temporal cluster) or spatial location (spatial cluster). The score of each supporting section is its height (raised to some power H) multiplied by its extent (raised to some power E). The E and H parameters therefore control the impact of cluster extent and height respectively.”

9) Did subjects with more contacts in a given region contribute more heavily to aggregate findings for that region?

We think this is less likely to be the case. For OFC, no single subject contributed to more than 7.23% of the contacts. The median fraction of contacts contributed by a single subject was 5.12% (range: from 0.6% to 7.23%).

Here we plot the subject-by-subject results in the OFC and present them in an order according to the number of electrode contacts each of them had.

This figure is now in the Supplement (Supplementary Fig. 21).

10) Is it coincidence that the three regions with the strongest value signals (OFC, hippocampus and insula) were also the three regions with the highest number of contacts?

We think this is possible, but we also see evidence against this conjecture. For example, the IPS had 62 contacts, and yet it did not show significant correlation with either the current or previous subjective value at the group level. By contrast, the PCC and amygdala had 31 and 30 contacts respectively, and in these two brain regions we found either the current subjective value (amygdala) or the previous subjective value (PCC) signals. Therefore, it is not entirely the case the regions with more contacts had the strongest value signals.

To discuss points (9) and (10) from the Reviewer, we add a paragraph in Discussion on page 35:

“Sparse and heterogeneous coverages also raise two important questions. First, is it possible that the three regions with the strongest value signals (OFC, hippocampus, and insula) were also the regions with the highest number of contacts? We think that while this is possible, we also saw

evidence against this conjecture. For example, the IPS had 62 contacts, and yet it did not show significant correlation with either the current or previous subjective value at the group level. By contrast, the PCC and amygdala had 31 and 30 contacts respectively, and in these two brain regions we found significant results on either the current subjective value (amygdala) or the previous subjective value (PCC). Therefore, it is not entirely the case that brain regions with more contacts had the strongest value signals. Second, is it possible that subjects with more contacts in a given region contribute more heavily to aggregate findings for that region? We think this is less likely to be the case. For OFC, no single subject contributed to more than 7.23% of the contacts. The median fraction of contacts contributed by a single subject was 5.12% (range: from 0.6% to 7.23%). To illustrate, in Supplementary Fig. 21, we plot the subject-by-subject results in the OFC and present them in an order according to the number of electrode contacts from each subject. In summary, we wish to point out that these remain important questions and need to be more systematically addressed as sEEG studies mature as a field.”

11) Typos:

- a. current subject value -> current subjective value
- b. Together, through -> Together,
- c. spilt/spit -> split
- d. the by the -> by the
- e. spare -> sparse
- f. fixation cross presenting -> fixation cross presented
- g. Subjects can take -> Subjects could take
- h. We did, however, observed -> We observed
- i. TFCE where we -> TFCE which we
- j. reverred -> reversed

Thank you. We now correct these typos and highlighted them in blue in the revised manuscript.

REVIEWER COMMENTS

Reviewer #1 (Remarks to the Author):

The authors did a great job addressing all my comments. I think the paper has been much improved and all the necessary control analyses have been conducted. Additionally, the authors have conducted a replication study in a healthy group of participants, which demonstrates that the neural data reported in this article can be considered as healthy signal given the similar behavior between patients and healthy controls.

Nevertheless, I still have a concern regarding my initial comment 3, which was:

“Overall, no statistics are reported in the results, making the reviewing procedure difficult. There is no effect size or t-statistics, and solely mentions of p-values lower than 0.05 and stars on the figures. The results are written as if they were a legend for all figures, without reporting enough details on the conducted analyses.[...]. The full results section is written in this form and should be re-written. »

Some statistics are now mentioned in the text, and supplementary figures for t-statistics have been included, but to me, statistics are still missing.

As an example (among ALL results): “Second, the willingness-to-pay in a trial was significantly affected by the willingness-to-pay in the previous trial” (which test, Statistics?). The larger the subjects’ bid in a trial, the higher she or he tended to bid in the next trial (which test, Statistics?) even though sequentially presented rewards were uncorrelated in preceding BDM-bid values (which test, Statistics?), indicating a temporal context dependency in bids.”

My requests for statistics are in brackets.

I leave it to the editor to decide how important it is to ask for this. To me, an article without clearly reported statistics is a no-go.

Nevertheless, I would be happy to see a revised version of this much improved and worthwhile article.

Reviewer #2 (Remarks to the Author):

The authors have responded well to my previous comments. A couple of minor points:

I would rephrase the sentence starting on line 222 to "However, for shorter response-time trials, this would include time points after the subjects had already made a decision and entered into the response phase of the trial"

The attempt to include more statistical precision appears to have led to include too many digits after the decimal point, e.g. for t-statistics - two seems enough to me

Reviewer #1 (Remarks to the Author):

The authors did a great job addressing all my comments. I think the paper has been much improved and all the necessary control analyses have been conducted. Additionally, the authors have conducted a replication study in a healthy group of participants, which demonstrates that the neural data reported in this article can be considered as healthy signal given the similar behavior between patients and healthy controls.

Nevertheless, I still have a concern regarding my initial comment 3, which was: “Overall, no statistics are reported in the results, making the reviewing procedure difficult. There is no effect size or t-statistics, and solely mentions of p-values lower than 0.05 and stars on the figures. The results are written as if they were a legend for all figures, without reporting enough details on the conducted analyses.[...]. The full results section is written in this form and should be re-written. »

Some statistics are now mentioned in the text, and supplementary figures for t-statistics have been included, but to me, statistics are still missing.

As an example (among ALL results): “Second, the willingness-to-pay in a trial was significantly affected by the willingness-to-pay in the previous trial” (which test, Statistics?). The larger the subjects’ bid in a trial, the higher she or he tended to bid in the next trial (which test, Statistics?) even though sequentially presented rewards were uncorrelated in preceding BDM-bid values (which test, Statistics?), indicating a temporal context dependency in bids.”

My requests for statistics are in brackets.

Thank you for the comments. We agree that the first revision could have been much more organized in its statistical presentation and have endeavored to include much more detail in this revision. We have now included descriptions of all of the statistical tests performed and their outcomes in the paper. Below we describe these changes separately for the behavioral analyses and the neural analyses.

For the behavioral analyses

We updated the following paragraph on page 8:

“We found several interesting features in the subjects’ willingness-to-pay. First, across all subjects, the distribution of willingness-to-pay appeared to be positively skewed (Fig. 1B). About 23% of the trials across all subjects were zero bids. Second, the willingness-to-pay in a trial was significantly affected by the willingness-to-pay in the previous trial: the larger the subjects’ bid in a trial, the higher she or he tended to bid in the next trial (Fig. 1C) even though sequentially presented rewards were uncorrelated in preceding BDM-bid values, indicating a temporal context dependency in bids. For each subject separately, we performed a linear regression analysis using current bid as data and bid from the previous trial as the regressor. To examine whether the regression coefficient is significantly different from 0, we performed a one-sample *t*-test on the regression coefficient ($\alpha = 0.05$, two-tailed). Across all subjects, the *t* statistics ranged from -1.72 to 5.37 (Supplementary Table 1 for reports on the *t* statistics). At the single-subject level, 10 out of 20 subjects showed a significant effect of the previous subjective

value (regression coefficients significantly different from 0 at $p < 0.05$ were marked with * in Fig. 1C). At the group level, the mean regression coefficient (across subjects) was significantly different from 0 (one-sample t test, $t = 4.77$, $p < 0.001$). Further analysis revealed that the bid in the trial presented two-trials back did not have a significant effect on the current bid (one-sample t test, $t = 0.41$, $p = 0.341$; see Supplementary Fig. 1 and Supplementary Table 2). To further examine whether such temporal context dependency can be considered normal, we ran the same task on 35 healthy subjects from the normal population and found that most subjects (25 out of 35) showed the same temporal context dependency, with the current bid positively correlating with the bid on the previous trial (one-sample t test, $t = 3.81$, $p < 0.001$; see Supplementary Fig. 2 and Supplementary Table 3; see also ref¹). Third, we found no relationship between response time (how long it took the subjects to place the bid) and the amount of their willingness-to-pay (Fig. 1D). For each subject, we performed a linear regression analysis using response time as data and willingness-to-pay as the regressor. A one-sample t test ($\alpha = 0.05$, two-tailed) was performed on the regression coefficient of willingness-to-pay. Across all subjects, the t statistics ranged from -1.63 to 10.20 (Supplementary Table 4 for reports on the t statistics). At the single-subject level, 4 out of 20 subjects showed significant effect of response time (regression coefficient significantly different from 0 at $p < 0.05$ were marked with * in Fig. 1D). At the group level, the mean regression coefficient of response time was not significantly different from 0 (one sample t test, $t = 1.60$, $p = 0.063$). We therefore concluded that there was no significant relation between response time and willingness-to-pay. The distribution of individual subjects' data on the willingness-to-pay and response time can be found in the Supplement (Supplementary Figs. 3 and 4). Individual subjects' scatterplots on the willingness-to-pay of the current trial against that of the previous trial and on the willingness-to-pay against response time can also be found in the Supplement (Supplementary Figs. 5 and 6)."

And also in the supplement:

Supplementary Figure 2

Supplementary Figure 2. Temporal context dependency of subjective value from 35 healthy subjects. For each subject, we regressed the subjective value (willingness-to-pay) of the current trial against that of the previous trial. Here we plot the mean regression coefficient (across subjects) of the previous subjective value. For each subject, we performed a one-sample t test ($\alpha = 0.05$, two-tailed) on the regression coefficient. Across all subjects, the t statistics ranged from -2.59 to 4.08. Regression coefficients significantly different from 0 at $p < 0.05$ were marked by the * symbol. The t statistic of these regression coefficients and its corresponding p -value can be found in Supplementary Table 3. We found that 25 out of 35 subjects showed positive regression coefficient with the previous subjective value. At the group level, the mean regression coefficient (across subjects) was significantly different from 0 (one-sample t test, $t = 3.81$, $p < 0.001$).

We added 4 tables in the Supplement (Supplementary Tables 1-4) to summarize the statistical results of the above behavioral analyses.

Supplementary Table 1: t statistic of regression coefficient of the previous SV in a linear regression analysis where the data were the current SV.

Subject	t statistic	p -value
1	3.16	0.002
2	2.21	0.028
3	2.63	0.009
4	4.63	<0.001
5	0.17	0.865
6	2.37	0.019
7	4.10	<0.001
8	-0.55	0.583
9	0.42	0.674
10	1.58	0.116
11	5.37	<0.001
12	1.81	0.071
13	-0.12	0.903
14	2.76	0.006
15	1.67	0.097

16	2.45	0.015
17	-1.72	0.087
18	3.57	<0.001
19	1.30	0.195
20	0.17	0.868

Supplementary Table 2: *t* statistic of regression coefficient of the previous trial (t-1) SV and two-trial back (t-2) SV where the current SV were the data in a linear regression analysis.

Subject	Previous trial (t-1)		Two-trials back (t-2)	
	t statistic	p -value	t statistic	p -value
1	2.96	0.004	-0.23	0.820
2	2.10	0.037	0.17	0.866
3	2.36	0.020	0.64	0.525
4	3.91	<0.001	1.14	0.256
5	0.17	0.869	0.15	0.878
6	2.03	0.044	1.54	0.126
7	3.28	0.001	0.64	0.521
8	-0.54	0.588	0.19	0.850
9	0.41	0.679	0.06	0.950
10	1.45	0.150	0.91	0.364
11	5.11	<0.001	-0.65	0.518
12	1.58	0.117	0.26	0.794
13	-0.36	0.717	1.10	0.271
14	2.57	0.011	0.11	0.910
15	1.87	0.064	-1.35	0.178
16	2.30	0.022	0.57	0.569
17	-1.83	0.069	-1.29	0.199
18	3.19	0.002	0.56	0.576
19	1.29	0.199	-0.01	0.994
20	0.36	0.719	-2.54	0.012

Supplementary Table 3: *t* statistic of regression coefficient of the previous SV where the current SV were the data from 35 healthy subjects in a linear regression analysis.

Subject	t statistic	p -value
1	1.47	0.144
2	-2.59	0.011
3	1.63	0.107
4	1.28	0.202
5	2.34	0.020
6	0.12	0.906
7	1.48	0.144
8	2.65	0.009
9	0.79	0.432
10	1.81	0.074
11	-1.60	0.113
12	0.65	0.515
13	-1.27	0.206
14	1.75	0.083
15	1.02	0.308
16	0.16	0.872
17	1.28	0.204
18	-0.08	0.937
19	3.22	0.002
20	0.36	0.721
21	1.57	0.119
22	1.93	0.056

23	4.08	<0.001
24	-0.33	0.742
25	1.74	0.085
26	-0.77	0.443
27	2.35	0.021
28	1.61	0.110
29	0.67	0.502
30	0.31	0.754
31	1.20	0.233
32	-0.02	0.982
33	-0.21	0.832
34	-0.46	0.643
35	-0.65	0.514

Supplementary Table 4: *t* statistic of regression coefficient of response time where the current SV were the data in a linear regression analysis.

Subject	t statistic	p -value
1	-0.49	0.626
2	0.52	0.606
3	0.73	0.467
4	0.92	0.358
5	3.06	0.003
6	0.45	0.655
7	-0.16	0.875
8	0.77	0.443
9	1.25	0.214
10	-1.63	0.105
11	0.01	0.994
12	-0.54	0.588
13	-1.20	0.232
14	2.36	0.020
15	1.02	0.307
16	2.02	0.045
17	10.20	<0.001
18	-0.49	0.625
19	0.06	0.954
20	-1.43	0.154

For the neural analyses

We did three things to strengthen our reports of the statistical tests:

1. We added a paragraph summarizing the statistical procedure in *Results*.
2. We added tables summarizing the statistical results of all group-level neural analyses (Table 1 & Supplementary Table 5-8).
3. We added a sentence describing the statistical tests after mentioning each neural result.

Below we describe each one of these 3 things we did in detail.

1. We note that since we used the same statistical procedure throughout (permutation test based on threshold-free-cluster-enhancement statistic), we described the statistical procedure in *Results* on page 10 and referred the reader to *Methods* for a detailed description:

“For all the neural GLM analyses, statistical significance was established based on a permutation test with threshold-free cluster enhancement (TFCE) as the test statistic^{64,65}. The TFCE statistic is used to describe signals that exhibit some spatial or temporal continuity without having to arbitrarily set a threshold for defining clusters of signals. For a given regressor of interest, we computed the TFCE statistic based on the z-statistic timeseries of the corresponding regression coefficients and for each time point separately. To correct for multiple testing across time points, a familywise error rate at 0.05 was implemented based on comparing the TFCE statistic at each time point with the null distribution of the maximum (or minimum) TFCE statistic (see *Group-level permutation test* in *Methods* for details). The time points that survived multiple-testing correction were marked with the * symbol (Figs. 2-8).”

2. Here we show the table added in the main text (Table 1). Table 1 includes details on the statistical results, from the test statistic, p-value, and information about the temporal cluster, presented separately for the current and previous subjective value. Table 1 (page 13 in the main text) includes all the brain regions we examined:

Table 1

ROI	Regressor	Cluster size	Start time (ms)	End time (ms)	Maximum or minimum TFCE	p-value	Peak time (ms)
OFC	Current SV	117	340	1500	17237135	<0.0001	710
	Previous SV	77	390	1150	-1477123	0.0003	1010
Medial OFC	Current SV	65	400	1040	857411	0.0039	690
	Previous SV	78	640	1410	-953291	0.0003	1020
Central OFC	Current SV	118	330	1500	11840031	<0.0001	850
	Previous SV	78	370	1140	-1090378	0.0013	890
Lateral OFC	Current SV	20	570	760	398460	0.0357	660
		16	1000	1150	403743	0.0353	1110
Anterior OFC	Current SV	136	150	1500	5448448	<0.0001	1010
		16	-670	-520	-371424	0.0442	-570
	Previous SV	61	-320	280	699254	0.0177	-220
		108	380	1450	-921352	0.0086	660
Posterior OFC	Current SV	75	360	1100	6125627	<0.0001	680
		20	1310	1500	373035	0.0298	1480
	Previous SV	80	410	1200	-2315647	0.0001	990
Amygdala	Current SV	15	560	700	218651	0.0357	590
Hippocampus	Current SV	9	-810	-730	-289468	0.0455	-770
		13	-690	-570	-368070	0.0308	-630
		29	20	300	-508621	0.0177	200
		18	360	530	-332990	0.0346	450
	Previous SV	63	290	910	-1943490	0.0003	760
Insula	Current SV	36	450	800	439686	0.0155	630
	Previous SV	23	1190	1410	238403	0.0277	1230
ACC & MCC	Previous SV	11	940	1040	-221955	0.0359	980
PCC	Previous SV	19	710	890	-207341	0.0314	850

Table 1. Summary of statistical analysis examining the effects of current and previous subjective value on high-gamma activity. Statistical significance was established based on a permutation test with threshold-free cluster enhancement (TFCE) as the test statistic. Details of the statistical procedure can be found in *Methods*. Here cluster size refers to the size of temporal cluster, i.e., number of consecutive time

points whose p -value are less than 0.05 after familywise error correction for multiple testing across time points. Since TFCE was calculated based on the z statistic, TFCE would be positive for positive effects, and negative for negative effects. The maximum TFCE indicates the strongest positive effect within the temporal cluster, while the minimum TFCE indicates strongest negative effect. The p -value was estimated based on the null distribution of the maximum TFCE statistic (for positive effects) or the null distribution of the minimum TFCE statistic (for negative effects) through permutations. The start and end time indicate the start and end time of the temporal cluster. Time at 0 indicates the onset of food stimulus presentation. The peak time corresponds to the time point with either the maximum or minimum TFCE within the temporal cluster.

We also added 4 tables summarizing the statistical results in the Supplement (Supplementary Table 5-8):

Supplementary Table 5: Summary of statistical analysis using different GLMs to examine the robustness of the effects of current and previous subjective value on high-gamma activity (80-150 Hz) in the OFC. Conventions are the same as Table 1 in the main text.

Model	Regressor	Cluster size	Start time (ms)	End time (ms)	Maximum or minimum TFCE	p -value	Peak time (ms)
GLM-2	Current SV	120	310	1500	14678969	<0.0001	710
	Previous SV	78	390	1160	-1556068	0.0006	990
GLM-3	Current SV	134	170	1500	12564742	<0.0001	700
	Previous SV	70	390	1080	-1319450	0.0008	490
GLM-4	Current SV	116	350	1500	16198243	<0.0001	700
	Previous SV	76	400	1150	-1589928	0.0004	680
	RT	17	440	600	-399944	0.0277	510
		21	700	900	-423796	0.0248	820
GLM-5	Current SV	117	340	1500	16336977	<0.0001	700
	Previous SV	78	410	1180	-1413531	0.0002	1020
GLM-6	Current SV	116	350	1500	17453375	<0.0001	680
	Previous SV	71	400	1100	-1565380	0.0003	680
	Power from previous trial	13	190	310	-173119	0.0205	230
		21	1300	1500	-304044	0.0034	1470
GLM-1 ITI = 1 s	Current SV	112	390	1500	3322871	0.0001	690
	Previous SV	8	980	1050	-223952	0.038	1010
GLM-1 ITI = 1.5 s	Current SV	117	340	1500	3364757	<0.0001	700
	Previous SV	79	670	1450	-541001	0.0074	980
GLM-1 ITI = 2 s	Current SV	79	280	1060	2505369	<0.0001	530
	Previous SV	63	420	1040	-1230608	0.002	610

Supplementary Table 6: OFC gamma activity (30-80 Hz). Summary of statistical analysis using different GLMs to examine the robustness of the effects of current and previous subjective value. Conventions are the same as Table 1 in the main text.

Model	Regressor	Cluster size	Start time (ms)	End time (ms)	Maximum or minimum TFCE	p -value	Peak time (ms)
GLM-1	Current SV	84	440	1270	16050565	<0.0001	770
	Previous SV	113	380	1500	-10108427	<0.0001	1030
GLM-2	Current SV	83	450	1270	12276075	<0.0001	800
	Previous SV	113	380	1500	-10478213	<0.0001	1020
GLM-3	Current SV	130	210	1500	13170589	<0.0001	790
	Previous SV	121	140	1340	-7699084	<0.0001	830
GLM-4	Current SV	82	450	1260	12923489	<0.0001	800
	Previous SV	112	390	1500	-9882367	<0.0001	1020
GLM-5	Current SV	84	440	1270	14950391	<0.0001	760

	Previous SV	112	390	1500	-10644560	<0.0001	1020
GLM-6	Current SV	6	-120	-70	-307970	0.0487	-80
		85	430	1270	13917136	<0.0001	770
	Previous SV	113	380	1500	-7624380	<0.0001	1020
	Power from previous trial	19	-600	-420	-204042	0.0202	-460
GLM-1 ITI = 1 s	Current SV	11	-960	-860	378495	0.0366	-910
		108	430	1500	4520680	<0.0001	800
	Previous SV	12	220	330	-312217	0.0419	290
GLM-1 ITI = 1.5 s	Previous SV	108	430	1500	-3321347	<0.0001	1280
GLM-1 ITI = 2 s	Current SV	7	-110	-50	-278142	0.0435	-70
		99	170	1150	3599589	<0.0001	790
	Previous SV	116	350	1500	-5418119	<0.0001	1050

Supplementary Table 7: OFC beta (13-30 Hz), alpha (8-12 Hz), and theta (4-7 Hz) activity. Summary of statistical analysis examining the effects of current and previous subjective value. Conventions are the same as Table 1 in the main text.

Frequency band	Regressor	Cluster size	Start time (ms)	End time (ms)	Maximum or minimum TFCE	p-value	Peak time (ms)
Beta	Current SV	16	-940	-790	439425	0.0285	-860
		12	-520	-410	-452762	0.0258	-460
	Previous SV	36	-780	-430	660197	0.0133	-580
		18	840	1010	385429	0.039	930
Alpha	Current SV	17	-470	-310	-449951	0.0259	-360
		22	-190	20	471555	0.0195	-140
		58	540	1110	-3227908	<0.0001	830
	Previous SV	20	-480	-290	570823	0.0158	-370
102		490	1500	18137892	<0.0001	960	
Theta	Current SV	5	-400	-360	-417332	0.0497	-380
		8	-150	-80	475021	0.0435	-110
		93	580	1500	-19634679	<0.0001	960
	Previous SV	28	-550	-280	692177	0.0261	-510
		110	490	1500	43945586	<0.0001	1340

Supplementary Table 8: OFC activity in the time-frequency space. Summary of statistical analysis examining the effects of current and previous subjective value. Conventions are the same as Table 1 in the main text.

Regressor	Cluster size	Start time (ms)	End time (ms)	Frequency range (Hz)	Maximum or minimum TFCE	p-value	Peak time (ms)	Peak frequency (Hz)
Current SV	4547	-70	1500	(22, 150)	1814219	<0.001	760	84
	460	560	1500	(4, 24)	-593651	<0.001	850	6
	140	1180	1500	(132, 150)	109157	0.015	1460	140
	117	-140	210	(32, 46)	-122188	0.019	-20	42
	50	-490	-290	(6, 12)	-119255	0.02	-330	6
	15	-510	-420	(14, 16)	-105159	0.037	-440	16
	8	510	580	(22, 22)	-96499	0.049	550	22
	8	540	590	(10, 12)	-106870	0.035	570	12
	3	1360	1380	(132, 132)	90233	0.041	1370	132
	2	1060	1070	(130, 130)	104626	0.019	1060	130
	1	410	410	(58, 58)	89793	0.042	410	58
	1	1210	1210	(62, 62)	124038	0.007	1210	62
Previous SV	1	1260	1260	(96, 96)	131017	0.006	1260	96
	3942	-40	1500	(20, 150)	-755778	<0.001	970	56
	859	230	1500	(4, 24)	1292569	<0.001	1360	4
	13	1420	1500	(90, 94)	-90208	0.043	1470	92
5	1340	1380	(104, 104)	-96185	0.03	1340	104	

	4	440	470	(128, 128)	-107898	0.022	460	128
	2	210	220	(48, 48)	-90484	0.042	220	48
	1	1100	1100	(132, 132)	-93380	0.037	1100	132

3. We included the following descriptions whenever we described a significant neural result:
“(permutation test, $p < 0.05$, familywise error corrected for multiple testing across time points).”

On page 14 before we report the robustness tests of subjective-value signals:

“The significant results reported below were based on permutation test using threshold-free-cluster-enhancement (TFCE) statistic as the test statistic ($p < 0.05$, familywise error corrected). The detailed statistical summary can be seen in Supplementary Tables 5 (for high-gamma activity) and 6 (for gamma activity). The t -statistic version of these results can be seen in Supplementary Fig. 13.”

Finally, at the end of each relevant figure (Figs. 2-8), we included the following descriptions on the statistical procedure:

“Colored (blue or green) horizontal lines with the * symbol on top or beneath indicate the time points with $p < 0.05$ (familywise error corrected) using permutation test with the threshold-free-cluster-enhancement (TFCE) statistic as the test statistic.”

I leave it to the editor to decide how important it is to ask for this. To me, an article without clearly reported statistics is a no-go.

Nevertheless, I would be happy to see a revised version of this much improved and worthwhile article.

Thank you for these valuable comments. As described above, we have now added information about the statistical tests and statistics to both the behavioral and neural analyses.

Reviewer #2 (Remarks to the Author):

The authors have responded well to my previous comments. A couple of minor points:

I would rephrase the sentence starting on line 222 to "However, for shorter response-time trials, this would include time points after the subjects had already made a decision and entered into the response phase of the trial"

Thank you. We rephrase the sentence on page 9 as the reviewer suggested.

The attempt to include more statistical precision appears to have led to include too many digits

after the decimal point, e.g. for t -statistics - two seems enough to me

Thank you. We now report t -statistics two digits after the decimal point on page 10:

“Across these OFC contacts, we found that high-gamma activity positively correlated with the current subjective value (maximum t statistic=7.02 at 710 ms after stimulus onset) but negatively correlated with the previous subjective value (minimum t statistic=-4.89 at 1020 ms after stimulus onset) (Fig. 2B; permutation test, $p<0.05$, familywise error corrected for multiple testing across time points; see Table 1 for reports on the statistics).”

REVIEWERS' COMMENTS

Reviewer #1 (Remarks to the Author):

The authors addressed my comments. I have no further comments.

REVIEWERS' COMMENTS

Reviewer #1 (Remarks to the Author):

The authors addressed my comments. I have no further comments.

Thank you for reviewing our paper and providing those detailed comments.